# Microhaplotype deep sequencing assays to capture *Plasmodium vivax* infection lineages

*Plasmodium vivax* elimination is challenged by dormant liver stages (hypnozoites) that can reactivate months after initial infection resulting in relapses. Relapsing infections confound antimalarial clinical efficacy trials due to the inability to distinguish between recurrences arising from blood-stage treatment failure (recrudescence), reinfection or relapse. Genetic relatedness of paired parasite isolates, measured by identity-by-descent (IBD), can provide important information on whether individuals have had single or multiple mosquito inoculations, thus informing on recurrence origin. We developed a high-throughput amplicon sequencing assay comprising 93 multi-SNP (microhaplotype) markers to determine IBD between *P. vivax* clinical isolates. The assay was evaluated in 745 global infections, including 128 infection pairs from a randomized controlled trial (RCT) (ClinicalTrials.gov NCT01680406). Simulations demonstrate low error in pairwise IBD estimation at the panel (RMSE < 0.12) and IBD-based networks illustrate strong clustering by geography. IBD analysis in the RCT demonstrates a lower frequency of suspected relapses or recrudescence in patients treated with primaquine compared to those without primaquine; the impact is greater when paired with chloroquine than with artemether-lumefantrine. Our results demonstrate the potential to derive new information on *P. vivax* treatment and transmission using IBD generated by amplicon sequencing data that can be further improved with time-to-event models.

Outside of sub-Saharan Africa, *Plasmodium vivax* is becoming the predominant cause of malaria[1]. The control and elimination of *P. vivax* is confounded by the parasite's ability to form dormant liver stages (hypnozoites) that can reactivate weeks to months after initial infection, causing recurrent episodes of malaria known as relapses[2]. Relapsing infections may enhance spatio-temporal *P. vivax* transmission and can cause repeated symptomatic illness, increasing the risk of anaemia and life-threatening disease[3]. The radical cure of vivax malaria requires treatment with both a schizontocidal antimalarial (chloroquine (CQ) or artemisinin combination therapy) combined with a hypnozoiticidal agent (primaquine (PQ) or tafenoquine (TQ))[1]. However, safe and effective treatment of hypnozoites is complicated by several factors. Both PQ and TQ can cause severe hemolysis in patients with a common enzymopathy, glucose-6-phosphate dehydrogenase

(G6PD) deficiency, and hence healthcare providers may be hesitant to provide these treatments in the absence of point-of-care tests to diagnose the condition. The risk of hemolysis and the efficacy of PQ and TQ regimens are determined in part by the total dose and length of the treatment regimen and may vary in different endemic settings[4–8]. Clinical trials of *P. vivax* with a long follow up period (generally 6-12 months) can provide critical insights on the risk of recurrence with different treatment regimens but are constrained in distinguishing whether recurrent infections are due to schizontocidal treatment failure (recrudescence), reactivation of hypnozoites (relapses), or a new mosquito inoculation (reinfection)[9,10].

Parasite genotyping of infection pairs (pre- and post-treatment), is well established for interpreting antimalarial clinical efficacy for *P. falciparum*, to distinguish between recrudescence (same/

✉ e-mail: Sarah.Auburn@Menzies.edu.au

homologous pairs) and reinfection (different/heterologous pairs)[11]. However, detecting *P. vivax* relapse events is complex as they that can be homologous or heterologous to the pre-treatment isolate[12–14]. Genomic studies of *P. vivax* have revealed that a proportion of paired clinical isolates that are classified as heterologous using traditional genotyping methods, can share homology in large segments of the genome, inferring recent common ancestry[15,16]. These familial relationships can be defined using genetic information on identity-by-descent (IBD). Closely related paired infections (e.g., siblings, with ≥50% IBD) are more likely to come from a single mosquito inoculation and thus to be relapses rather than reinfections. Genome-wide data are ideal for quantifying IBD. However, this is not feasible in most malaria-endemic settings. Targeted next generation sequencing (NGS)-based methods, such as amplicon sequencing, offer an affordable and versatile alternative[17]. Previously we established a bioinformatic framework for retrieving genome-wide panels of *P. vivax* microhaplotype markers; short (200 bp) genomic segments comprising multiple high diversity single nucleotide polymorphisms (SNPs)[18]. Using in-silico analyses, we demonstrated that ~100 *P. vivax* microhaplotypes can reliably capture IBD between paired isolates and could be used to define spatio-temporal patterns of *P. vivax* infection[18].

In this study, we advanced our previous in-silico work by designing an NGS-based amplicon sequencing assay for 98 *P. vivax* amplicons, including 93 microhaplotypes that capture IBD accurately in diverse endemic areas. Our assay was designed to primarily target clinical (symptomatic) cases. Our rhAmpSeq library preparation protocol incorporates all markers in a single plex reaction and was able to generate high throughput, cost-effective, sensitive and specific data from global isolates. Standard population-level genetic metrics could be applied to the data to highlight potential use cases for National Malaria Control Programs (NMCPs) that could inform *P. vivax* treatment options and transmission dynamics. Proof of concept was provided in the assessment of radical-cure efficacy in a primaquine trial conducted in Ethiopia.

## Results

### A rhAmpSeq multiplex of 93 microhaplotypes with high sensitivity and specificity

A two-step multiplex PCR-based library preparation assay was established using rhAmpSeq chemistry (Integrated DNA Technologies, IDT). The multiplex assay amplifies one species confirmation marker, four putative markers of drug resistance, and 93 microhaplotypes distributed relatively uniformly across the *P. vivax* genome (Supplementary Data 1; Supplementary Fig. 1). To evaluate the assays technical and analytical performance, a total of 805 *P. vivax* infections were assessed, including 128 paired *P. vivax* infections from a randomized controlled trial, two *P. vivax* serial dilution panels, and 15 non-*P. vivax* control samples (Supplementary Data 2, Supplementary Fig. 2).

The specificity of the assay was evaluated using a selection of 8 non-vivax *Plasmodium* spp. and 4 uninfected human DNA controls. Examination of amplicon coverage revealed that 86 markers (89% [86/97] after excluding the mitochondrial species marker) exhibited amplicon coverage <0.9 in all negative controls (Supplementary Fig. 3a). Amongst the 11 markers with coverage >0.9 in one or more negative controls, 6 had read counts <25 in all controls tested, with only 5 markers exceeding 25 reads in the *P. knowlesi* controls (Supplementary Fig. 3b).

The assay sensitivity was assessed using serial dilutions of two *P. vivax*-infected patient blood samples (KV3 and KV5) under high sample multiplexing (library pooled across 384 samples), using both the standard PCR step 1 DNA input and reaction volumes (referred to as full chemistry with a 20 µl reaction volume) and halving the PCR step 1 DNA input and reaction volumes (referred to as half chemistry with a 10 µl reaction volume). Under both full and half chemistry conditions, >84 (87%) markers were successfully genotyped (>25

reads) in the 70 and 96 parasite/ul sample preparations, as well as in higher densities (Supplementary Figs. 4a and b). Further details on read counts can be found in Supplementary Figs. 4c–f.

Parasite density defined by microscopic blood film examination may not correlate directly with parasite DNA yield owing to the presence of multiple life cycle stages in *P. vivax* infections (schizonts, for example, having greater DNA quantity than ring stages). To address this, we evaluated the threshold cycle (Ct) values of the KV5 serial dilution using real-time PCR, targeting the *pvmtcox1* gene. Samples with a Ct <30 (KV5 96 parasite/ul) had successful genotyping at 87 (89.6%) markers (Supplementary Fig. 5, Supplementary Table 1).

Our serial dilutions were prepared using blood collected into anticoagulant-coated tubes, that generally yield higher quality DNA than dried blood spots. We determined how the number of successfully genotyped markers correlated with parasitemia in dried blood spot samples from a study in Ethiopia. There was a weak correlation between parasitemia and read count (rho = 0.1873, $p$ = 0.0217); 148 cases (98.67%) with parasitemia ranging from 240 to 42,280 parasites/µl could be genotyped successfully at ≥80 (86%) microhaplotype markers (Supplementary Fig. 6).

### Geographically diverse application potential

A total of 745 *P. vivax* isolates were used to evaluate amplification efficacy from 8 countries across the globe. Successful genotyping was defined as ≥25 reads for >80% of the 97 nuclear genome amplicons and could be achieved in 705 (94.6%) isolates. 564 (83%) independent (non-recurrent) samples were taken forward for evaluation of individual marker efficacy and country-specific amplification patterns. Aside from 248 samples, data from all isolates was generated from a single 384-well run (run 3). Four microhaplotype markers (64,721, 354,590, 419,038 and 466,426) exhibited lower read-pair than read amount counts but none failed consistently across populations (Fig. 1b).

### Accuracy in *P. vivax* variant calling

Using a set of 110 isolates with both whole genome sequencing (WGS) and amplicon sequencing data, the accuracy of SNP variant calling (applying the default 10% minor allele threshold, minor allele depth of 2, and minimum depth of 5 and 25 for WGS and amplicon sequencing data respectively) was confirmed at the 425 SNPs within the 97 markers (only excluding mitochondria). Concordance was observed at 96.4% (42,805/44,386) of the genotype calls (Supplementary Table 2, Supplementary Data 3). Homozygous reference versus homozygous alternate allele discordances contributed 0.33% (148/44,386) of all calls. Most discordant calls reflected heterozygous versus homozygous calls (90.6% (1433/1581)), likely reflecting differences in the limit of genotyping of minor clones between the datasets. After applying a 1% minor allele threshold to both datasets, a notable difference was more than doubling of heterozygous amplicon sequencing calls that were homozygous in the WGS dataset (from 227 calls with 10% threshold to 697 with 1% threshold; Supplementary Table 3). As illustrated in Supplementary Fig. 7, a large proportion of discordant heterozygote genotype calls appear to reflect a greater limit of genotyping in the deep sequencing (amplicon) data.

### High potential to capture within-host infection complexity

Using a set of 104 isolates with both WGS and microhaplotype data, within-host infection complexity was explored using the $F_{ws}$ and effective multiplicity of infection (eMOI). As illustrated in Fig. 2a, b, there was a strong correlation between the genome-wide $F_{ws}$ and the microhaplotype-based eMOI in each geographic region assessed (Spearman's rank correlation, rho = −0.6109985, $p$ = 5.664e-12) (Supplementary Data 4).

The eMOI distributions in Fig. 2c illustrate trends in within-host diversity at the country level in a larger panel of 562 isolates from 6 populations which had over 30 isolates available: Afghanistan ($n$ = 157),

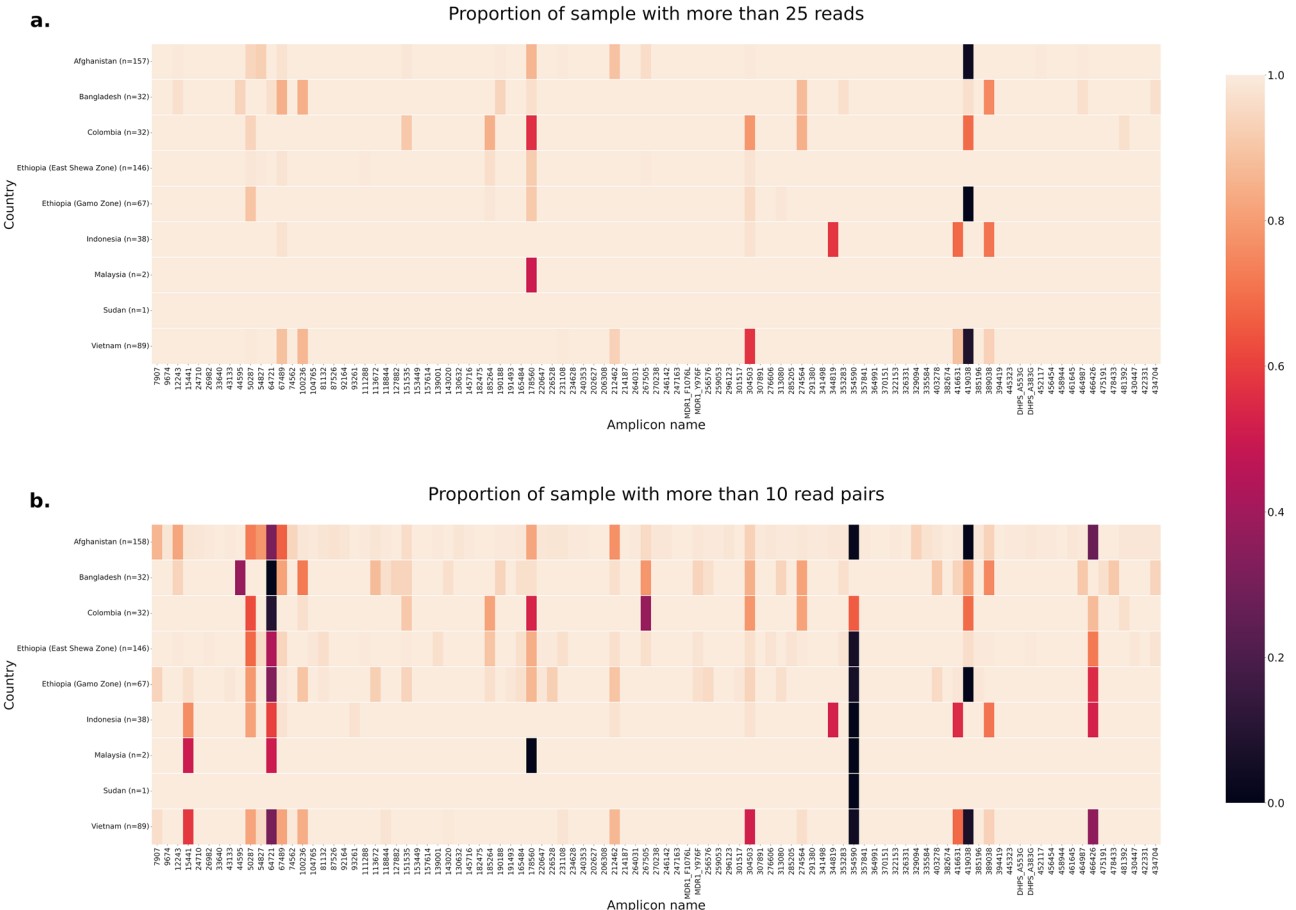

**Fig. 1 | *P. vivax* proportion of samples with more than 25 reads or more than 10 read pairs by country.** Heat maps illustrate the proportion of samples with more than 25 reads (**a**) and proportion of samples with more than 10 read pairs derived from *DADA2* (**b**) for each marker by country. Markers (x-axis) were ordered by chromosome and coordinate. All samples with genotype failures ( < 25 reads or <10 read pairs) are presented in black. Microhaplotype markers 64721, 354590, 419038 and 466426 displayed consistently low read-pair counts. Data are presented on a

total of 564 independent (not including replicates or recurrences) infections that passed genotyping. The data from Ethiopia are split into dried blood spots (DBS) from a clinical trial conducted in East Shewa Zone and whole blood (WB) extracts from a therapeutic efficacy survey conducted in Gamo Zone for comparative assessment. The DBS versus WB comparisons reveal similar read and read pair counts at all loci.

Bangladesh ($n = 32$), Colombia ($n = 32$), Indonesia ($n = 38$), Ethiopia ($n = 214$), and Vietnam ($n = 89$). The lowest eMOI distribution was observed in Sumatra, Indonesia, suggestive of low endemicity relative to the other sites. Provincial distributions are provided in Supplementary Fig. 8.

### Effective IBD capture

To assess the ability of the microhaplotype panel to capture IBD accurately, we used *paneljudge* (an R package to judge the performance of a panel of genetic markers using simulated data) to simulate the relative mean square error (RMSE) in estimation of nine pairwise IBD states: IBD = 0.01, 0.05, 0.1, 0.15, 0.2, 0.25, 0.5, 0.75 and 0.99. The simulations were run in the 6 major populations using data from 91 assayable microhaplotypes (markers 354,590 and 419,038 excluded), revealing moderately high diversity (mean diversity range 0.44–0.67: Supplementary Fig. 9) and low RMSE ( < 0.12 for all pairwise IBD states) in each population (Supplementary Fig. 10, Fig. 3).

### Potential of microhaplotype-based IBD to inform on recurrence

To demonstrate the potential of the assay to inform on the origin of recurrences, the microhaplotype-based IBD was determined using *DCifer* on data from 128 pairs of initial and recurrent infections (Supplementary Data 5). These isolates came from individuals enrolled into a randomized controlled trial (RCT) at two sites in

Ethiopia and treated with either chloroquine (CQ), CQ plus primaquine (PQ), artemether-lumefantrine (AL) or AL plus PQ with one-year follow-up to assess the efficacy of PQ regimens for radical cure[19]. In the original RCT, genotyping data at 1–7 microsatellites were generated in 46 pairs of infections in which recurrence occurred within 42 days of follow-up. The microsatellite number was too low to enable accurate IBD determination; hence, pairs were defined as heterologous (reinfection/relapse) or homologous (recrudescence/ relapse). Of the 46 pairs, 34 had microhaplotype data from our study. Amongst the 34 pairs, 44% (15/34) were defined as heterologous using the microsatellite data compared to 18% (6/34) defined as strangers (arbitrary threshold of IBD < 25%) with the microhaplotype-based IBD estimate (Supplementary Data 5). The relative over-estimation of strangers with microsatellite data underscores the improved insight from IBD analysis.

Using the full set of 128 microhaplotype-genotyped pairs from the RCT, we used IBD distributions and thresholds to inform on probable relapse/recrudescence versus reinfection risks in each treatment arm under the assumption that highly related pairs are more likely to reflect relapses than reinfections (assumption suitable for low-inbreeding populations). Our results demonstrate that patients treated without PQ had a higher median IBD (1.0, IQR 0.66–1.0) compared to those treated with PQ (0.50, IQR 0.05–0.99) ($p = 0.004$, Mann-Whitney U test), consistent with a greater risk of relapsing and recrudescent

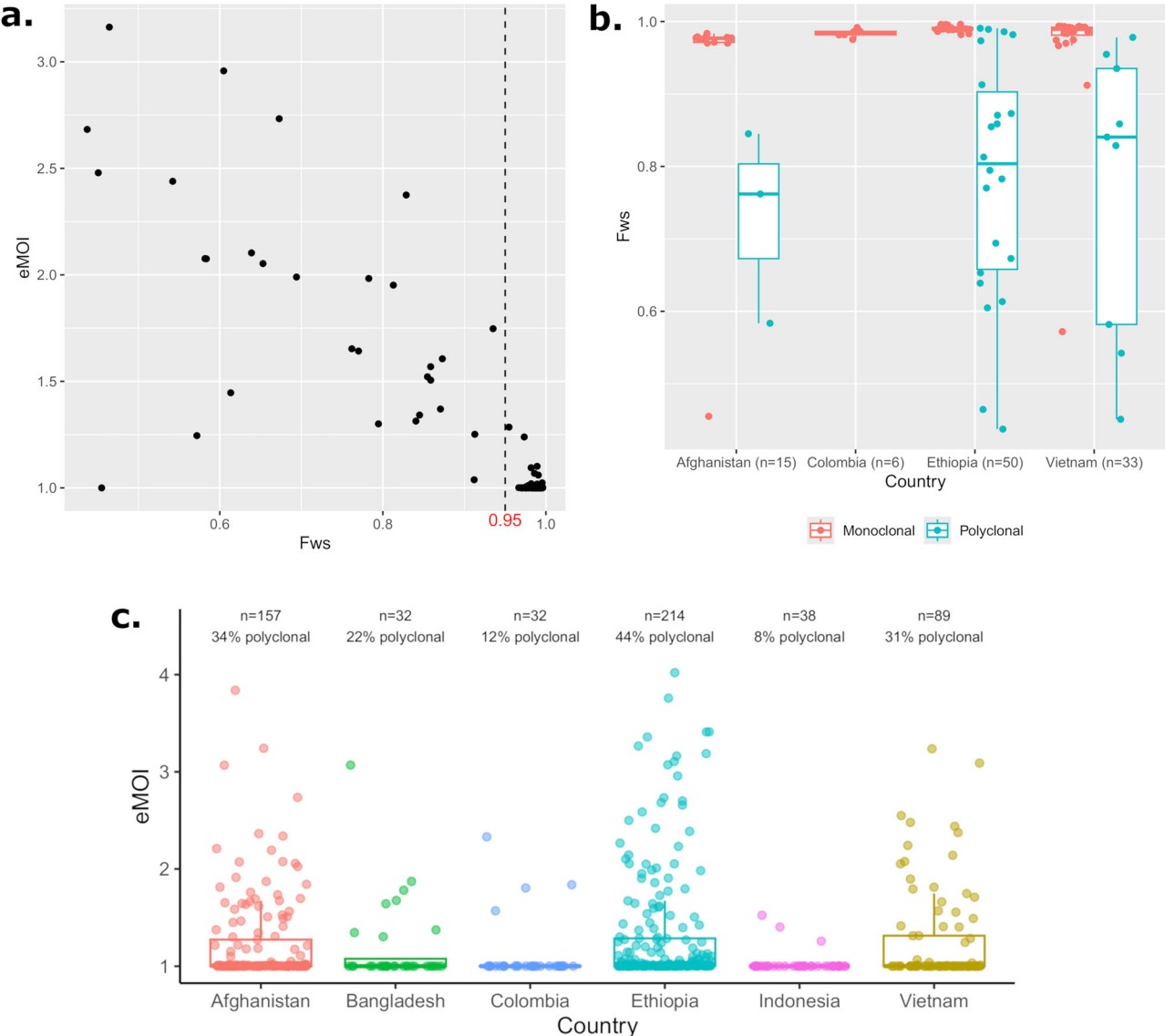

**Fig. 2 | Microhaplotype-based within-host diversity trends. a, b** illustrate the level of concordance between genomic (as measured by the Fws) and microhaplotype (as measured by eMOI) data in estimation of within-host *P. vivax* diversity in 104 independent cases. The boxplots present the median, interquartile range and minimum and maximum value. Overall, high concordance is observed between the two datasets. **c** presents the eMOI distributions at the country using data on 562 independent cases. The boxplots present the median, interquartile range and minimum and maximum value. Each country is presented in a different colour.

infections. In the CQ arm, 4% (2/46) of genotyped recurrences occurred by day 28, inferring majority of highly related pairs were likely relapses. The median IBD in the CQ arm (0.98, IQR 0.57–1) was significantly higher than the CQ + PQ arm (0.36, IQR 0–0.98) ($p = 0.008$, Mann-Whitney U test) (Fig. 4a). Using the IBD ≥ 25% threshold, a higher proportion of highly related pairs were observed in the CQ (84%, 38/45) versus CQ + PQ arm (54%, 7/13) ($\chi2 = 3.8$, $p = 0.051$).

In the AL arm, 41% (23/56) of recurrences occurred by day 42 (range 21–42 days) with less clear distinction of relapse versus recrudescence. The AL arm had higher IBD (1, IQR 0.71–1) than the AL + PQ arm (0.86, IQR 0.26–1) but the difference was not as large as observed with CQ and was not statistically significant ($p = 0.155$, Mann-Whitney U test) (Fig. 4a). There was also no significant difference between the AL (91%, 51/56) and AL + PQ (71%, 10/14) arms with the IBD ≥ 25% threshold ($\chi2 = 2.3$, $p = 0.129$). Owing to the differences in PQ impact amongst the CQ and AL arms, we compared the median IBD of the CQ + PQ (0.36, IQR 0–0.98) and AL + PQ (0.86, IQR 0.26–1) arms, but the difference was not significant ($p = 0.116$, Mann-Whitney U test).

We assessed the distribution of IBD over time using a 120-day threshold in line with the typical relapse periodicity of 4 months for African *P. vivax* infections[20]. Recurrences within 120 days of treatment had higher median IBD (0.98, IQR 0.58–1.0) compared to those after 120 days (0.73, IQR 0.12–1.0). The results are consistent with a rising proportion of reinfections at later time points, but the difference was not significant ($p = 0.083$, Mann-Whitney U test) (Fig. 4b). Using the IBD ≥ 25% threshold, significantly more highly related infections were observed pre (89%, 11/96) versus post (66%, 21/32) day 120 ($\chi2 = 7.32$, $p = 0.007$).

**Spatial transmission potential**

The utility of the marker panel to define spatial transmission dynamics was explored using the amplicon sequencing data and genotypes derived from whole genome sequencing data. As illustrated in the neighbour-joining tree in Fig. 5, SNP-based identity-by-state (IBS) analyses on 728 low-complexity infections demonstrated a high degree of differentiation by geographic location. In areas with both

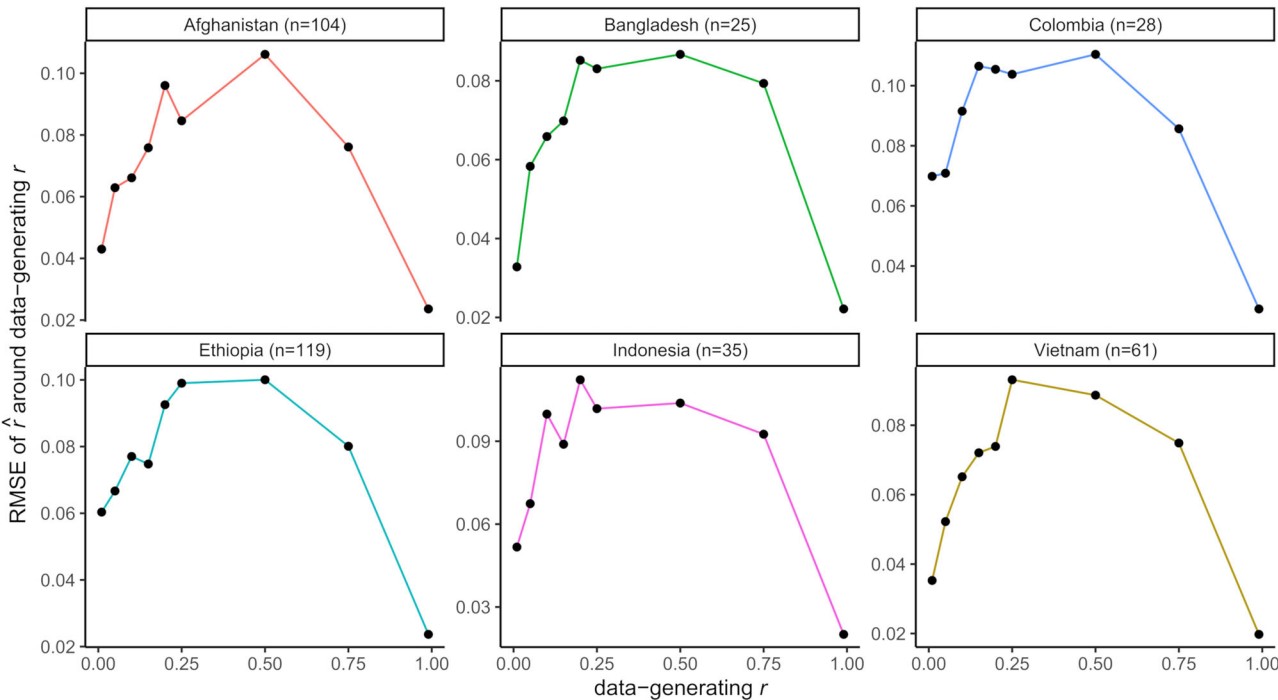

**Fig. 3 | Simulations of IBD estimation in different geographic areas using the microhaplotype panel.** Root mean square error (RMSE) of relatedness estimates based on data simulated using nine different data-generating relatedness estimates, $r$ (specifically IBD of 0.01 [essentially unrelated], 0.05 [very low related], 0.1 [low related], 0.15 [low related], 0.2 [low related], 0.25 [half-sibling], 0.5 [sibling], 0.75 [highly related], and 0.99 [essentially clonally identical]) with switch rate

parameter $k$ set to 5. Data were generated using *paneljudge* software on 91 high performance microhaplotypes in independent infections from each population (see sample sizes within the figure). In all populations, half-siblings and siblings had the highest RMSE, but this remained below 0.12 in all cases. Each country is presented in a different colour.

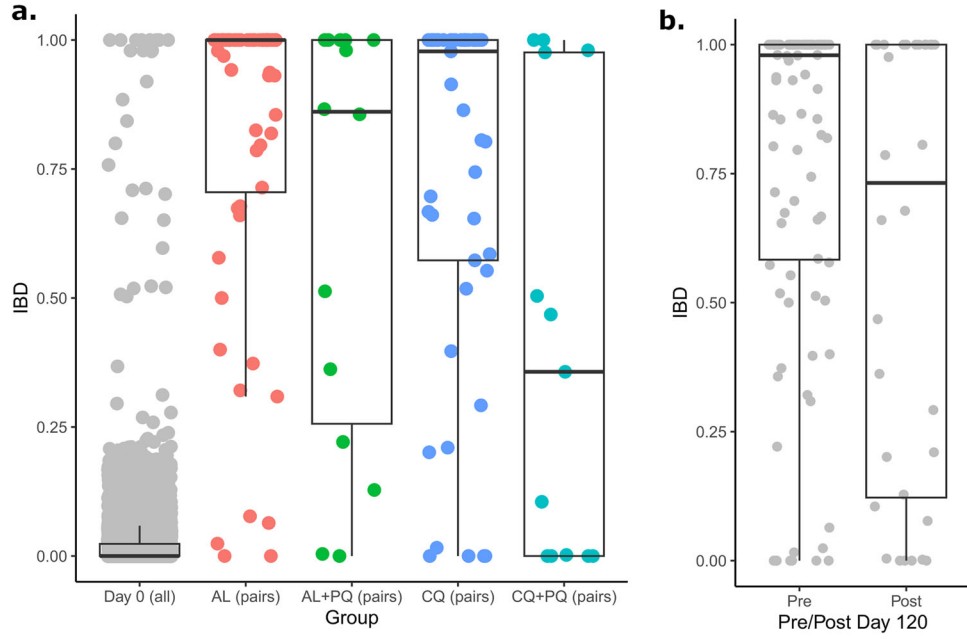

**Fig. 4 | IBD distributions by treatment and time to recurrence in a randomized controlled trial conducted in Ethiopia. a** presents the IBD distributions in initial and recurrent infection pairs across all pairs and grouped by treatment arm; AL (Artemether-Lumefantrine), CQ (Chloroquine), AL + PQ (AL + Primaquine) and CQ + PQ. Each treatment is presented with a different colour. **b** presents the same IBD data grouped by recurrences occurring less versus more than 120 days after the initial infection. Each boxplot presents the median, interquartile range and

minimum and maximum value. Data are presented on recurrence pairs from 128 independent patients, with only one infection pair presented per patient to avoid potential bias ($n$ = 128 patients, 256 samples). Majority (90%, 115/128) of pairs reflect day 0 and recurrence 1 time points. Where the day 0 or recurrence 1 infections failed genotyping or had inconclusive clinical metadata, consecutive pairs of recurrence 2 to 4 pairs were used instead as the patients received the same treatment up to recurrence 4.

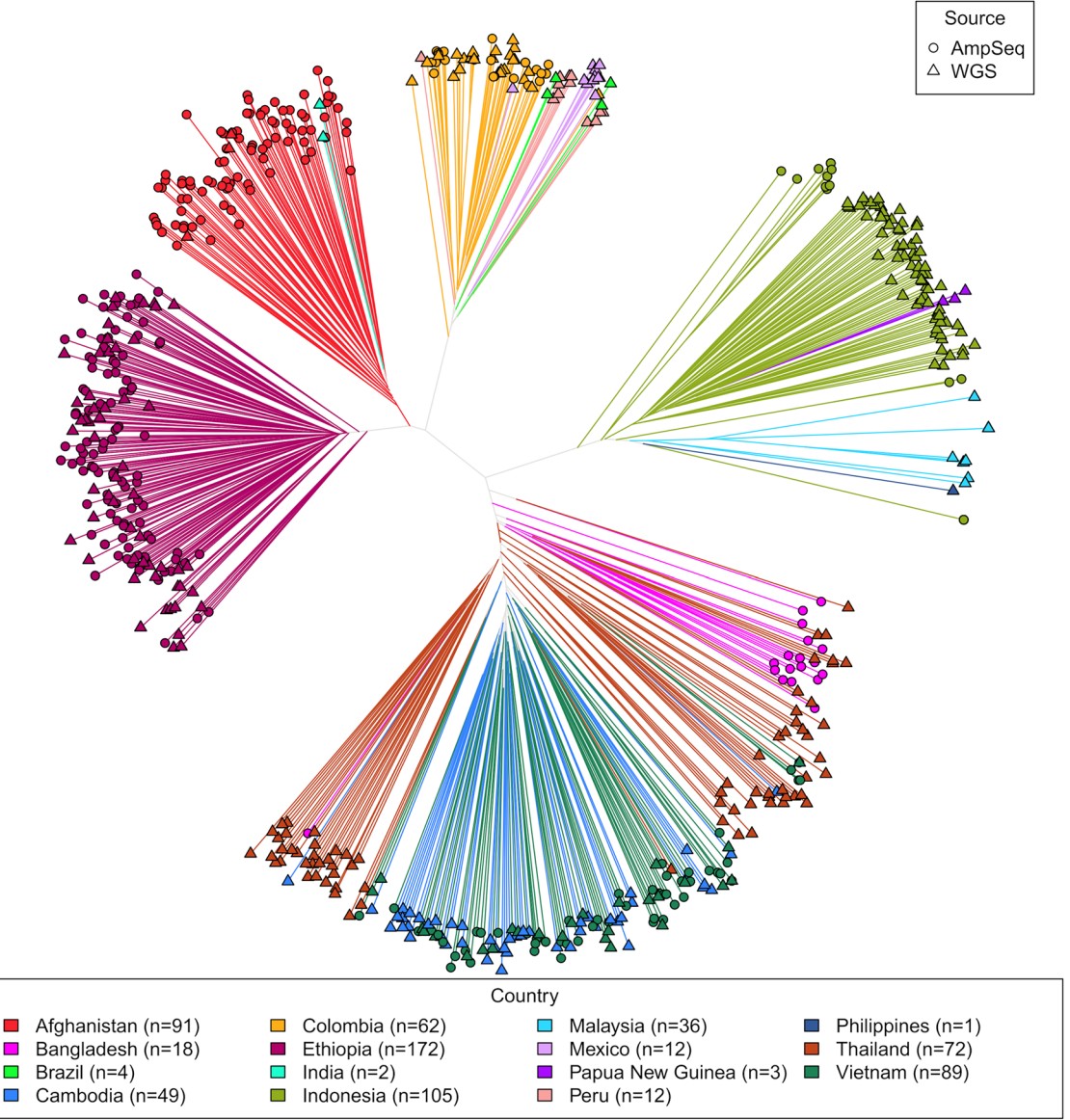

**Fig. 5 | Identity-by-state-based spatial patterns using AmpSeq and WGS data.** The plot presents an unrooted neighbour-joining tree derived from a distance matrix on the microhaplotype calls using genotypes derived from microhaplotype and WGS data at the microhaplotype marker positions only. The neighbour-joining tree illustrates largely distinct clustering by country, except for neighboring Cambodia and Vietnam, and Bangladesh and Thailand. In countries with both WGS (triangles) and AmpSeq data (circles), no evidence of clustering by sequencing method is observed; although separation is observed in Indonesia, the WGS data from this region derives from Papua Province, whilst the AmpSeq data derives from Sumatra Province, located on different islands nearly 2500 Km apart. The plots were generated using data on 728 independent, monoclonal infections.

amplicon sequencing and genomic data, there was no evident differentiation by methodology, supporting the robustness of merging the data sources. IBD analyses, which rely on accurate population allele frequency estimates, were focused on the amplicon sequencing data from the large sample sets ($n \geq 30$) and used both clonal and polyclonal infections. The IBD analyses revealed evidence of local and inter-site transmission networks even at IBD thresholds exceeding ~25% (Fig. 6). For example, moderate differentiation was observed between Gia Lai and Binh Phuoc Province in Vietnam, and between Gamo Zone and East Shewa Zone in Ethiopia (Fig. 6d, f). Supplementary Fig. 10 illustrates the connectivity patterns at thresholds as low as 5%, further demonstrating the differentiation within Ethiopia and Vietnam. IBD analysis also revealed clonal clustering in Sumatra, consistent with the low eMOI distribution (Fig. 6e).

## Effective plasmodium species confirmation

In addition to the microhaplotypes, a previously described mitochondrial amplicon was included in the assay to confirm *Plasmodium* spp.[21]. The assay amplifies coordinates PvP01_MIT_V2:2904-3149, which include species-specific SNPs and indels. Using *P. vivax* samples from a range of countries, and *P. falciparum*, *P. malariae*, *P. ovale* spp. and *P. knowlesi* negative controls, we confirmed amplification of the mitochondrial marker at moderate to high depth and coverage in all *Plasmodium* species (range 28–5236) (Supplementary Table 4). The method also allows for detection of mixed-species infection, which was tested in 3 artificially mixed samples with *P. vivax* and *P. falciparum*. Concordance between PCR-based and mitochondrial species classification was confirmed for each of the *Plasmodium* samples, with 2/3 artificially mixed samples detected successfully.

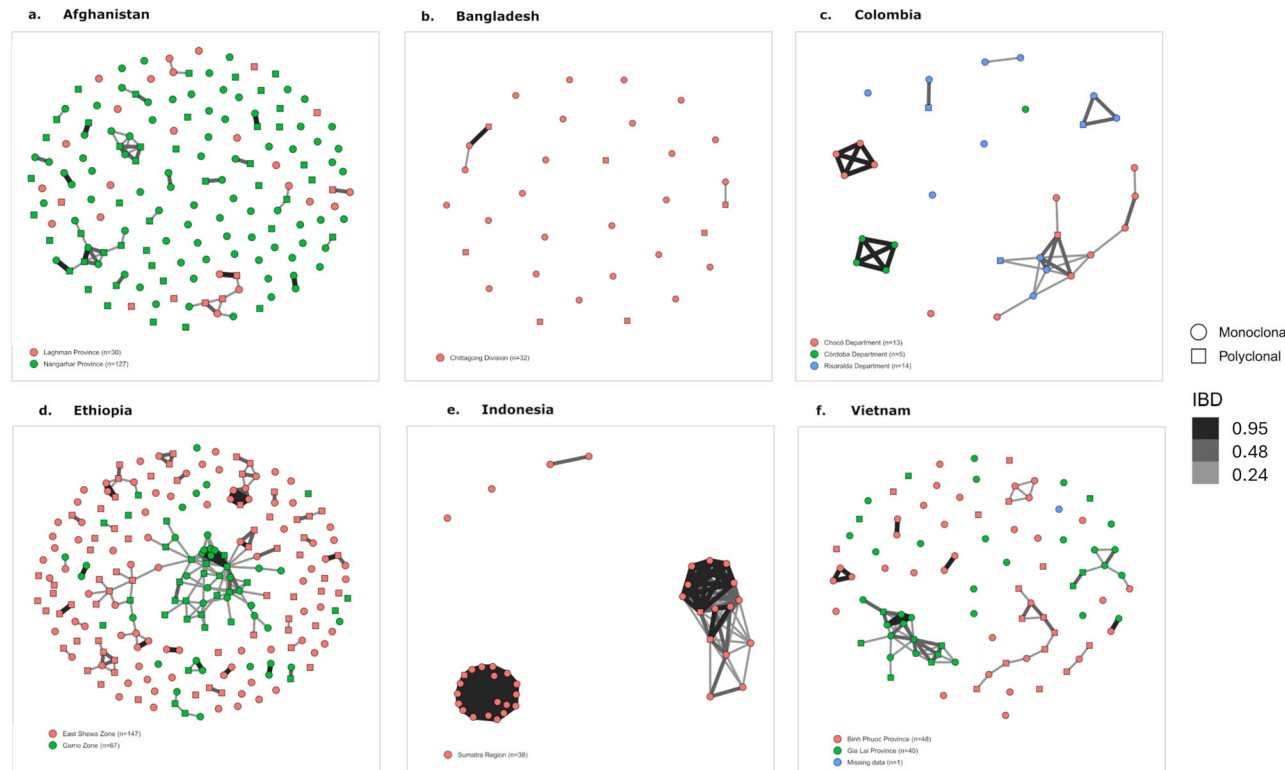

**Fig. 6 | IBD-based spatial patterns using microhaplotype data. a–f** present networks illustrating IBD-based connectivity between infections in Afghanistan (**a**), Bangladesh (**b**), Colombia (**c**), Ethiopia (**d**), Indonesia (**e**) and Vietnam (**f**). Each shape reflects an infection, colour-coded by site, and with shapes reflecting monoclonal (circle) versus polyclonal (square) infections. For each country, connectivity (illustrated by connecting lines on a grey scale between shapes) is presented at IBD thresholds ranging from ≥0.24 (thin, light grey lines) to ≥0.95 (thick, black lines). The baseline sample positions are based on the ~25% (0.24) IBD output. IBD measures were calculated on the microhaplotype calls using *DCifer* software. At the ~25% IBD threshold, large networks (10 or more connected infections) are largely confined to cases from the same site, with one or two cases connecting cases or other networks from different sites; these networks shrink with increasing IBD in all sites except for Sumatra, Indonesia, where the networks appear to reflect clonal clusters (retained at IBD ≥ 95%). All plots were generated using data on independent infections with sample sizes shown within the plots.

## Drug resistance candidates

Our assay included a non-exhaustive selection of amplicons encompassing candidate markers of antimalarial drug resistance including the multidrug resistance 1 (*pvmdr1*) 976 and 1076 loci, and dihydropteroate synthase (*pvdhps*) 383 and 553 loci that have been associated with ex vivo or clinical phenotypes (see ref.[22]). The dihydrofolate reductase (*pvdhfr*) 57, 58, 61 and 117 loci did not multiplex effectively with the other markers based on bioinformatic predictions and were thus not included, which limits detection of full sulphadoxine-pyrimethamine resistance. The prevalence of the variants by population is summarized in Fig. 7a and Supplementary Table 5. The prevalence of the *pvmdr1* Y976F variant, the most widely characterized candidate of CQR, ranged from 0% in Afghanistan to 100% in Sumatra, Indonesia[23]. The F1076L variant, which has also been implicated in CQR, exceeded 90% frequency in all countries except Colombia (4%). The prevalence of A383G mutation in *pvdhps*, a marker of antifolate resistance, varied highly, ranging from 2% in Afghanistan to 86% in Colombia. The *pvdhps* A553G mutation was observed at 3% in Vietnam and 20% in Bangladesh but absent in other populations.

## Discussion

We have established a highly multiplexed rhAmpSeq assay (up to 98 amplicons) to address critical knowledge gaps in our understanding of antimalarial efficacy and transmission dynamics of *P. vivax*. Our assay can be applied to low-density infections and to NGS platforms that are generally available in reference laboratories in malaria-endemic settings.

A critical requirement of the assay was its ability to be implemented in surveillance frameworks in vivax-endemic countries. This necessitated high sensitivity to genotype low-density, symptomatic *P. vivax* infections at affordable cost using locally accessible sequencing platforms. In a meta-analysis of 27 *P. vivax* clinical studies that did not apply a parasitemia threshold for enrolment, 95% of febrile patients had parasitemia >120 parasites per microliter blood[24]. With full chemistry, our analyses of serial dilutions demonstrated that >80% genotyping was successful when parasite density was >70 per microliter whole blood. Hence, we estimate that >95% of symptomatic *P. vivax* cases should be successfully genotyped. Defining a meaningful assay sensitivity with amplicon sequencing approaches is challenging as several factors other than parasite density can impact on read depth, including the yield of the sequencing platform used, the level of sample multiplexing, the relative target biomass of other samples on the same run, and whether a pre-amplification step is applied. To stress-test our assay, we evaluated the sensitivity of target detection under conditions at the lower end of potential sequence yield at the benefit of lower processing cost. Sensitivity evaluations were undertaken on a MiSeq platform, which is the most widely available platform in malaria endemic settings. The paired end read yield per run is ~20–30 million for the MiSeq compared to 260–800 million and up to 40 billion on the NextSeq and NovaSeq platforms, opening the possibility for even greater yield if the setting permits. Furthermore, samples could be pooled and did not require pre-amplification with selective Whole Genome Amplification (sWGA) or Primer Extension Preamplification (PEP)[25,26]. Our rationale was that the rhAmpSeq RNase H2 enzyme-dependent amplicon sequencing chemistry should reduce

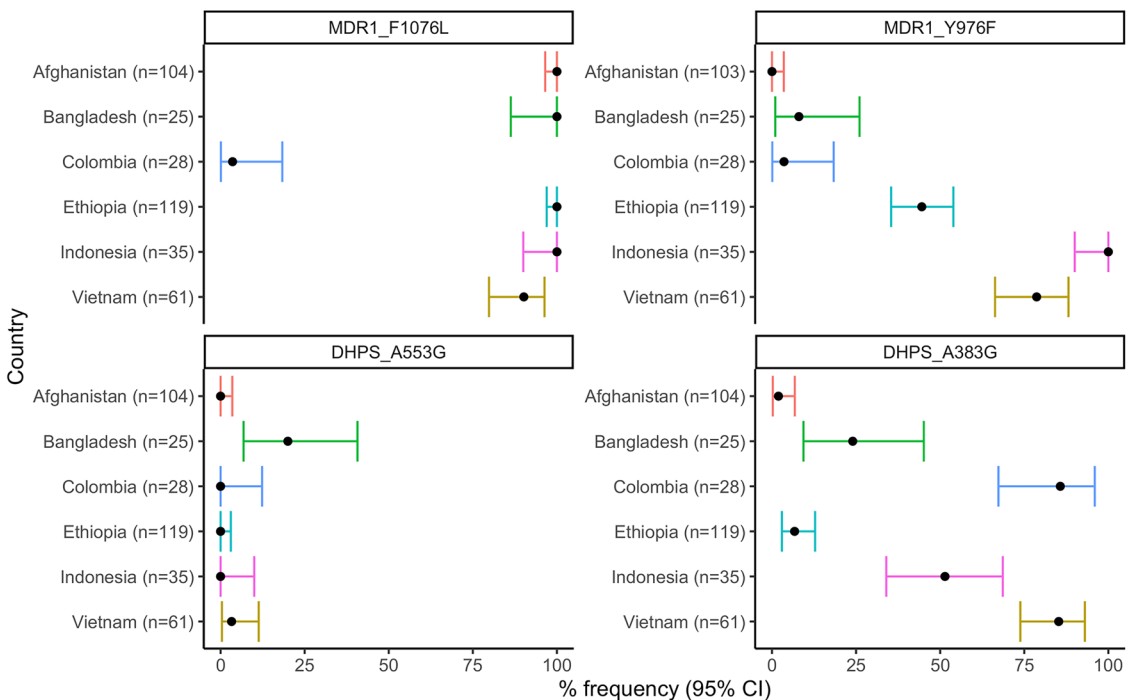

**Fig. 7 | Amino acid frequencies at *P. vivax* drug resistance candidates.** The plots present frequency and corresponding upper and lower 95% confidence intervals (CIs) for the given amino acid changes in baseline population samples from each country. All frequencies reflect the suspected drug resistance-conferring amino acid. All plots were generated using independent, monoclonal samples (*n* = 372). Each country is presented in a different colour.

the need for pre-amplification steps relative to standard AmpSeq as it minimizes spurious primer-primer interactions and off-target amplification[27].

Another essential data analysis requirement of our assay was the ability to generate genetic data that effectively capture pairwise IBD in *P. vivax* infections with the aim of defining the origin of recurrent infections, particularly for the detection of relapses. Previously described *P. vivax* amplicon sequencing assays, such as a widely used 42-SNP barcode, have not comprised the density of SNPs needed to capture IBD accurately[18,28,29]. Previously using in-silico methods, we demonstrated that panels of ~100 *P. vivax* microhaplotypes can capture pairwise IBD states ranging from 0 (unrelated) to 1 (clonally identical) with low error (RMSE under 0.12)[18]. In our current study, we applied the same in-silico models to demonstrate that, despite less optimal spacing and slightly fewer markers than in the in-silico panels, our assayable set of microhaplotypes has comparably low error in IBD estimation (RMSE under 0.12 with 91 microhaplotypes). We also note the adaptability of the rhAmpSeq protocol, enabling addition of new markers, which could be selected to cover genomic regions with low marker density. The software developed in our previous study supports microhaplotype selection that can be customized to specific populations or for specific marker traits[18].

In addition to the in-silico analysis, we explored microhaplotype-based IBD patterns between pre and post treatment peripheral blood isolates collected from a clinical trial conducted in Ethiopia[19]. The trial evaluated the safety and efficacy of combining PQ with either CQ or AL for the radical cure of *P. vivax*. There was a 2- to 3-fold reduction in the risk of recurrence within 12 months compared to patients treated with CQ or AL monotherapy, but there was no significant difference in the risk of recurrence between the CQ + PQ and AL + PQ arms. In line with the clinical data, our genetic analysis of 128 independent pairs of infections demonstrated a higher proportion of highly related (IBD ≥ 0.25) paired isolates in patients not treated with PQ. Some of the highly related pairs may reflect recrudescence but we postulate that the majority reflect relapses in the absence of radical cure. At the time

of the study (2012-14), there was no local evidence for AL resistance even in *P. falciparum*. There was evidence of low-grade CQ resistance in *P. vivax*, although only 4% of the genotyped recurrences in the study occurred before day 28. The integration of genetic models of clonal versus inter-episode sibling relationships with time-to-event analyses will eventually support further dissection of relapses from recrudescence[30]. The genetic data provided additional insights to interpret the clinical trial. Whilst there was no overall difference in the risk of recurrence between patients treated with PQ + CQ and PQ + AL, PQ + CQ reduced the proportion of highly related pairs compared to PQ + AL. This result may reflect CQ's slower elimination and longer post-treatment prophylaxis compared to AL, that suppressed early relapses[31]. Furthermore, CQ has been shown to have synergistic activity with PQ, that is not apparent with AL, and this may have had greater suppression of relapses and related recurrences[32]. However, the difference between CQ + PQ and AL + PQ did not reach significance: further studies are warranted to explore this. Lastly, we observed a greater frequency of paired infections with low relatedness (IBD < 0.25) after 4 months consistent with an increasing risk of reinfections during prolonged follow-up after antimalarial treatment. Our findings contribute proof of concept on the utility of genetics to inform on recurrence and highlight the potential insights that can be gleaned from recurrence classification of clinical trials. However, there are areas needing further research and development. Data from known relapse events, i.e. from individuals who were relocated to non-endemic areas, will be needed to further enhance knowledge of IBD distributions without confounding by reinfection. Further work is also needed to dissect the potential contribution of the splenic reservoir to recurrent *P. vivax* infections[33]. Lastly, as well as integrating time-to-event information and resolving inter-sibling and inter-clone parasites, modeling approaches that include information on endemicity as a prior will be important. For example, the high inbreeding in Sumatra would complicate distinction of relapses from reinfections in this low endemic setting where many infections from independent patients are highly related. However, with low endemicity, it also follows that the

risk of reinfection should be low. Our data have been made open access to support further modeling developments.

The microhaplotype data produced in this study also provides evidence of the potential for our assay to inform spatial patterns of diversity and connectivity. At a macro-epidemiological level, there was a distinct separation of populations by national boundaries in most areas, that could facilitate detection of cross-border importation of infection. Areas of the Vietnam-Cambodia border are impossible to separate, even with whole genome sequencing data, owing to extensive cross-border movement[34]. Within countries, we observed evidence of moderate to large local transmission networks, largely confined to specific sites but with some connections across sites. In Vietnam, for example, differentiation was observed between Binh Phuoc and Gia Lai Province, located ~300 Km apart. As the data repositories grow, it should be possible to better define the major transmission networks within local regions and their epidemiological context to inform on major reservoirs of infection and routes of infection spread between communities. The patterns of within-host diversity and connectivity revealed differences in transmission dynamics at the country and province level. These dynamics may not be readily captured by standard measures of parasite incidence as relapses are a major attributor to the *P. vivax* burden. Amongst the sites, Sumatra, Indonesia, had notably low prevalence of polyclonal infections and large clonal clusters supported low transmission and subsequent inbreeding in Sumatra relative to the other sites. The observed patterns infer that Sumatra is approaching the pre-elimination phase and may be receptive to targeted rather than broad scale intervention approaches.

Our panel included a non-exhaustive selection of *P. vivax* drug resistance candidates. The markers and mechanisms of resistance to the more widely used drugs such as chloroquine, artemisinin and partner drugs are not well understood in *P. vivax*[35]. The insights on treatment failure that can be made on the available markers, including those incorporated in our panel, is therefore limited from an NMCP perspective. However, the adaptability of the rhAmpSeq protocol ensures that new drug resistance markers can be readily added to the panel as they arise. Insights from the current selection of candidates included notable variation between countries in the prevalence of the *pvmdr1* Y976F variant, which has been implicated as a minor modulator of chloroquine (CQ) resistance[23]. The Y976F variant was present at 100% frequency in Sumatra, Indonesia. To our knowledge, this is the first report on Y976F in Sumatra, and aligns with the high prevalence (100%) of this variant in Papua province, Indonesia, a region that has historically harbored high-grade CQ resistant *P. vivax* infections[23,36,37]. Although not at fixation, high prevalence of the Y976F variant was also observed in Vietnam (79%) and Ethiopia (45%), but not in Afghanistan (0%). Therapeutic efficacy surveys report low level (<10%) CQ resistance in Bangladesh, Colombia, Ethiopia, Vietnam and Afghanistan, while 16-65% failures by day 28 have been reported in Sumatra, Indonesia[38–51]. The connection between the Y976F variant and clinical CQ efficacy is unclear but further exploration of phenotypic associations utilizing IBD and time-to-event data should lend further insight.

The panel also includes a previously described mitochondrial amplicon, comprising SNPs and indels that support *Plasmodium* spp. determination[21]. We used the amplicon to successfully confirm *Plasmodium* spp. in a selection of non-*P. vivax* controls. A caveat of the assay is the inability to distinguish *P. ovale curtisi* from *P. ovale wallikeri*. However, this could be addressed with the addition of other mitochondrial amplicons[52]. Although most assays displayed high specificity to *P. vivax*, several primers yielded amplicons of *P. knowlesi* DNA with high coverage (>0.9) and moderate to high read depth, but this species is rare outside of Malaysian and Indonesian Borneo.

Other targeted *P. vivax* genotyping assays are available, with designs tailored to different use cases[53]. Several barcodes with amplicons targeting 40–200 single SNPs have been designed to capture infection diversity and inform on transmission but were not specifically designed to measure IBD, which requires consideration of both marker polymorphism and spacing[28,54,55]. Most panels have been designed for amplicon sequencing but a recent panel of over 1200 SNP targets was designed for genotyping using molecular inversion probes (MIPs)[53]. The MIP panel was not evaluated for IBD determination, but the large number of markers brings potential. However, large marker numbers also confer relatively lower per marker sequence depth and less flexibility in sample throughput. MIP typing is not yet being used in malaria-endemic countries where amplicon sequencing offers a nimbler tool for surveillance. In terms of costs, our assay is comparable to other methods, with library preparation kits costing approximately $AUD 13–15 per sample (for 1000 to 10,000 reactions) when using half-chemistry reaction volumes.

In summary, the tools generated in our study provide a major inroad to establish high-throughput genetic data on *P. vivax* at reasonable cost that can provide policy relevant information on transmission dynamics, parasite reservoirs and the spread of infection across national and provincial borders.

## Methods

### Marker selection and assay design

193 microhaplotype markers, 5 drug resistance candidates, and two mitochondrial *Plasmodium* species-confirmation markers, were selected for assay design (Supplementary Data 1). The set of 193 microhaplotypes were derived from two 100-microhaplotype panels (referred to as the high-diversity and the random microhaplotype panels respectively based on the SNP filtering methods) that we previously demonstrated exhibit high accuracy in IBD determination[18]. The rationale for selecting two panels was based on expectation that some markers in the preferred panel (the high-diversity panel) might not be assayable. The drug resistance candidates were not chosen to reflect an exhaustive list of resistance candidates but rather a selection of SNPs that have previously been associated with ex vivo or clinical phenotypes[22]. The species-confirmation markers have been described in a previous malaria amplicon panel[21]. Primers and multiplex pools were designed by Integrated DNA Technologies (IDT). Primer specifications included non-mapping to the human (GRCh38.p14), *P. falciparum* (Pf3D7), *P. malariae* (PmUG01), *P. ovale wallikeri/curtisi* (GH01), and *P. knowlesi* (PKNH) reference genomes. A variant calling format (VCF) file for the open-access MalariaGEN Pv4.0 dataset, comprising 911,901 high-quality variants at 1895 global *P. vivax* samples, was also provided to confirm non-mapping to highly variable regions of the *P. vivax* genome[56]. A set of 148 markers that met the described primer specifications, and compatible within a single-plex reaction, were taken forward for a pilot experimental run, comprising 176 positive and 16 negative controls (Supplementary Data 1). From the pilot data (not presented here), a set of 98 markers were selected for the final assay (Supplementary Data 1). Requirements for the final panel included high specificity to *P. vivax* and approximate uniform microhaplotype distribution across the genome (Supplementary Fig. 1).

### Patient samples

The sensitivity and specificity of the 98-plex assay were evaluated using *P. vivax*-infected serial blood dilutions and unmodified *P. vivax*-infected blood samples, stored as dried blood spots or refrigerated whole blood in EDTA-coated microtainer or vacutainer tubes. The dilution series was prepared by mixing each of the three *P. vivax*-infected whole blood samples (microscopy-determined parasite densities of 740–960 parasites/μL blood) with uninfected human whole blood at 10-fold serial dilutions (0.7–0.96 parasites/μL blood). Whole blood samples from uninfected humans (*n* = 4), *P. falciparum* (*n* = 2), *P. malariae* (*n* = 2), *P. ovale* (*n* = 1), and *P. knowlesi* (*n* = 3) were included as negative controls. The samples were collected within the framework of previously described clinical trials and cross-sectional

surveys, as well as returning travelers presenting with malaria at the Royal Darwin Hospital, Australia, and represented 8 vivax-endemic countries (Supplementary Data 2)[19,47,57,58]. For recurrence classification, genotyping was conducted in samples from a randomized controlled trial (RCT) of radical cure regimens conducted in Ethiopia (ClinicalTrials.gov NCT01680406), for which the primary outcomes of the RCT have been published elsewhere[19]. DNA extraction was undertaken using Qiagen's QIAamp kits for respective dried blood spots or whole blood.

## Real-time PCR

The cycle threshold (Ct) values of *P. vivax* DNA samples were assessed using an Applied Biosystems® QuantStudio™ 6 Flex Real-Time PCR System (ThermoFisher Scientific) with a TaqMan real-time PCR-based assay, as adapted from ref.[59]. This assay detects *P. vivax* by targeting a conserved region of the mitochondrial cytochrome oxidase 1 gene (*pvmtcox1*)[59]. Each qPCR reaction mixture consisted of 10 μL TaqMan Universal Mix II master mix (ThermoFisher Scientific), 1.6 μL of both the forward and reverse *Pv-mtcox1* primers (10 μM), 0.8 μL of the *pvmtcox1* probe (10 μM), 2 μL H2O, and 4 μL gDNA. The thermocycling conditions included an initial denaturation at 95 °C for 10 min, followed by 45 cycles of denaturation at 95 °C for 15 s, and annealing and elongation at 60 °C for 1 min. Data analysis was performed using QuantStudio RT-PCR software to derive the threshold cycle (Ct) values.

## Whole genome sequencing (WGS) data

A combination of newly generated and pre-existing WGS data was used in the study (Supplementary Data 6). The pre-existing WGS data were derived from the MalariaGEN Pv4.0 repository[56]. Briefly, the data were generated using 100–150 bp paired-end sequencing on a HiSeq or MiSeq instrument (Illumina). The resultant reads were analyzed with the biallelic SNP pipeline used in this study, which incorporated read mapping against the *P. vivax* P01 reference using BWA-MEM2 and variant calling using the Genome Analysis Toolkit v4.5[60,61]. A total of 911,901 high-quality biallelic SNP loci were derived. Positions with less than 5 reads were considered genotype failures, with a minimum of two reference and two alternate alleles, and a minimum of 10% of minor allele read frequency required to define a genotype as heterozygous. The new WGS data were generated using microscopy-positive *P. vivax* infections collected within a therapeutic efficacy survey of chloroquine, conducted in Arbaminch, Ethiopia, in 2019. Briefly, DNA was extracted from 1–2 ml white blood cell-depleted (Plasmodipur-filtered) whole blood samples using Qiagen's QIAamp blood midi kits. Library preparation and sequencing were conducted on the gDNA at the Wellcome Sanger Institute using the Illumina HiSeq platform, generating 150 bp paired reads. The resultant reads were mapped against the PvP01 reference genome and genotype calling was conducted using the 911,901 MalariaGEN Pv4.0 SNPs, following the same methods. Only samples with more than 50% genotype calls were included in the analysis.

## rhAmpSeq library preparation and sequencing

Library preparation of the 98-plex marker panel was conducted using IDT's rhAmpSeq methodology. Details on the protocol are provided in Supplementary Note 1 and on the protocols.io website (https://www.protocols.io/private/B941F93F1F7D11F0BADE0A58A9FEAC02). In brief, genomic DNA samples were subject to a single multiplexed (all 98 targets amplified within one pool) primer amplification using pre-balanced, customized, rhAmpSeq primers. The product was diluted and subject to a second PCR, incorporating indexes and Illumina sequencing adaptors. The products from the second PCR step were pooled (incorporating up to 384 samples), resulting in the final library. The library was bead purified, assessed for quality (fragment size and quantity), and sequenced with 150 bp paired-end clusters on a MiSeq instrument (Illumina).

## Amplicon data read mapping and variant calling

Options for variant calling from fastq files at both biallelic SNPs and multiallelic microhaplotypes were incorporated into the data processing pipeline, which is available at https://github.com/vivaxgen/MicroHaps and described in Supplementary Note 2. In short, for the biallelic SNP pipeline, VCFs are generated using a pipeline that conducts read mapping via *bwa-mem2*, and variant calling with GATK v4.5. Positions with less than 25 reads were considered genotype failures, with a minimum of two reference and two alternate alleles, and a minimum of 10% of minor allele read frequency required to define a genotype as heterozygous. For the microhaplotype-calling pipeline, we adapted an existing *P. falciparum* microhaplotype pipeline which utilizes the *DADA2* software for denoising[62]. The microhaplotype pipeline includes adapter and primer removal with *cutadapt*, and custom scripts to postprocess *DADA2* output.

## Population genetic measures

For the resulting microhaplotype data, within-host infection diversity was characterized by the effective multiplicity of infection (eMOI) metric using *MOIRE* software (burnin=200, sample_per_chain=1000, pt_chains=40)[63]. MOIRE also generated information on the probability that each sample was polyclonal. Samples with >50% probability were categorized as polyclonal infections; this categorization was used when selecting monoclonal infections to perform analyses such as *paneljudge* simulations or IBS-based measures. For the WGS data, within-host diversity was characterized using the $F_{WS}$ score, calculated from biallelic SNP loci[64]. The R-based *paneljudge* package was used to assess the relative mean square error (RMSE) of the microhaplotype markers at IBD (relatedness, *r*) levels ranging from 0 (unrelated) to 1.0 (identical). Details of the equations, assumptions and parameters can be found on GitHub (https://github.com/aimeertaylor/paneljudge). For each country, we estimated allele frequencies for the markers in the microhaplotype panel. These allele frequency estimates were then entered into *paneljudge* to evaluate the panel's per-country capacity to generate data for accurate relatedness estimation. In brief, for each *r* value evaluated, data on 100 parasite pairs were simulated under the hidden Markov model HMM model, with inter-marker distance computed by the difference between the amplicon (marker) midpoints. For each pair, simulated data were then used to estimate *r* (using *paneljudge* (i.e., under the same model used to simulate the data). The RMSE of the 100 estimates around the data-simulating *r* value was then computed. Absolute rather than relative RMSE values were chosen as large relative errors at very low relatedness values are inconsequential for most analyses, particularly of recurrence where the focus in on pairs above a threshold (e.g., relatedness approximating 0.25). Between-sample IBD was measured using *DCifer*[65]. Network plots were generated using the *R*-based *ggnetwork* package (https://github.com/briatte/ggnetwork). Identity-by-state (IBS)-based measures of genetic distance were calculated using the *R*-based ape package, treating the microhaplotypes as multiallelic variants, and presented as a neighbour-joining tree plot[66].

## Plasmodium speciation based on pairwise genetic distance

To confirm the *Plasmodium* spp., amplicons targeting a universal mitochondrial sequence region were mapped to a synthetic genome. The synthetic genome was constructed using reference genomes of five *Plasmodium* species: PvP01_MIT_v2:2904-3149 (*P. vivax*), PmUG01_MIT_v1:1115-1361 (*P. malariae*), PKNH_MIT_v2:1632-1875 (*P. knowlesi*), Pf3D7_MIT_v3:1634-1877 (*P. falciparum*), and PocGH01_MIT_v2:2894-3137 (*P. ovale* spp.). The synthetic genome was generated through an alignment containing the specified regions within the reference genomes, aligned using *MUSCLE* v5[67].

The reads from each sample were mapped to the synthetic genome. The mappings were used to guide the merging of read-pairs. Each

merged read-pair that has less than 20% non-standard bases (due to missingness) was then compared pairwise to each sequence within the multi-sequence alignment containing all the reference genomes. The species of the sample was determined based on the highest pairwise alignment score.

## Inclusion and ethics

The authors in this study combine researchers from the malaria-endemic countries represented in the study (Afghanistan, Bangladesh, Colombia, Ethiopia, Indonesia and Vietnam) and nonmalaria endemic areas. The researchers from malaria-endemic countries were involved throughout the research process to ensure that the assays are implementable and impactful in local communities affected by malaria. All samples in this study were derived from blood samples obtained from patients positive for malaria and collected with informed consent from the patient, or patient's parent/legal guardian where individuals were less than 18 years of age. At each location, sample collection was approved by the appropriate local institutional ethics committees. The following committees gave ethical approval for the partner studies: Human Research Ethics Committee of NT Department of Health and Families and Menzies School of Health Research, Darwin, Australia; Islamic Republic of Afghanistan Ministry of Public Health Institutional Review Board, Afghanistan; ICDDR,B Ethical Review Committee, Bangladesh Comite Instiucional de Etica de Investigaciones en Humanos, Colombia; Comite de Bioetica Instituto de Investigaciones Medicas Facultad de Medicina Universidad de Antioquia, Colombia; Armauer Hansen Research Institute Institutional Review Board, Ethiopia; Addis Ababa University College of Natural Sciences, Ethiopia; Addis Ababa University, Aklilu Lemma Institute of Pathobiology Institutional Review Board, Ethiopia; National Research Ethics Review Committee of Ethiopia; Eijkman Institute Research Ethics Committee, Jakarta, Indonesia; Research Review Committee of the Institute for Medical Research and the Medical Research Ethics Committee (MREC), Ministry of Health, Malaysia; Scientific and Ethical Committee of the Hospital for Tropical Diseases in Ho Chi Minh City, Vietnam; The Ministry of Health Evaluation Committee on Ethics in Biomedical Research, Vietnam.

## Reporting summary

Further information on research design is available in the Nature Portfolio Reporting Summary linked to this article.

## Data availability

The microhaplotype-based parasite reads for all high-quality (pass) data generated in this study have been deposited in the European Nucleotide Archive under project ID PRJEB89933. The MalariaGEN Pv4.0 WGS data are available through the European Nucleotide Archive[56]. The WGS reads from the newly generated, high-quality Ethiopian *P. vivax* infections are available through the European Nucleotide Archive. The ENA accession numbers for all WGS samples used here are listed in Supplementary Data 6.

## Code availability

The pipelines for retrieving SNP data in variant call format (VCF), as well as microhaplotype data in Compact Idiosyncratic Gapped Alignment Report (CIGAR) string format, are available on GitHub at https://github.com/vivaxgen/MicroHaps[68].

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

## Acknowledgements

This work was supported, in whole or in part, by the Bill & Melinda Gates Foundation INV-043618 (S.A.). Under the grant conditions of the Foundation, a Creative Commons Attribution 4.0 Generic License has already been assigned to the Author Accepted Manuscript version that might arise from this submission. The study was also supported by the National Health and Medical Research Council of Australia (APP2001083 awarded to S.A.). The whole genome sequencing component of the study was supported by the Medical Research Council and UK Department for International Development (award number M006212 to D.K.) and the Wellcome Trust (award numbers 206194 and 204911 to D.K.). The OPRA clinical trial was supported by the Bill & Melinda Gates Foundation (OPP1054404 awarded to R.N.P.). We thank the patients who contributed their samples to the study, and the health workers and field teams who assisted with the sample collections. Whole genome sequencing was undertaken by the Wellcome Sanger Institute, and amplicon sequencing was undertaken at the Australian Genome Research Facility (AGRF) and Menzies School of Health Research. We thank the staff of the Wellcome Sanger Institute and AGRF for contributions to samples logistics, sequencing and informatics. We thank Ruchit Panjal for his assistance with the pipeline setup.

## Author contributions

S.A., D.N. and R.N.P. conceived the study. S.A., M.K., A Rumaseb, E.S., and H.T. designed the study. A Rumaseb, T.P., D.H., G.W., S.V.S, N.T.T.N. and S.A. contributed to the laboratory assay design. A Rumaseb, T.P., D.H. and A Rai contributed to genotyping data generation. S.V.S., R.D.P., R.A. and D.P.K. contributed whole genome sequencing data production and informatics support. M.K., H.T., K.S.H., A.O., P.M. and D.N. contributed to bioinformatic pipeline development. M.K., E.S., H.T., K.S.H. and S.A. conducted data analysis. E.D.B., N.T.T.N., N.H.C., A.A., T.S.D., D.T.A., A.G.R., A.P.P., I.S., M.S.A., Z.P., T.L.-M., D.E., T.W., N.M.A., M.J.G., N.P.D., N.J.W., A.R.T., D.P.K. and R.N. contributed essential field-based malaria collections and metadata, or guidance on the study design and interpretation.

## Competing interests

The authors declare no competing interests.

## Additional information

**Mariana Kleinecke**[1,24], **Edwin Sutanto** [2,3,24], **Angela Rumaseb**[1], **Kian Soon Hoon** [1], **Hidayat Trimarsanto**[1,4], **Ashley Osborne** [1], **Paulo Manrique**[5,6], **Trent Peters**[7], **David Hawkes**[7], **Ernest Diez Benavente** [8], **Georgia Whitton**[9], **Sasha V. Siegel** [9], **Richard D. Pearson** [9], **Roberto Amato** [9], **Anjana Rai**[1], **Nguyen Thanh Thuy Nhien** [10], **Hoang Chau Nguyen** [10], **Ashenafi Assefa** [11], **Tamiru S. Degaga**[12], **Dagimawie Tadesse Abate** [12], **Awab Ghulam Rahim** [13], **Ayodhia Pitaloka Pasaribu**[14], **Inge Sutanto**[15], **Mohammad Shafiul Alam** [16], **Zuleima Pava**[1], **Tatiana Lopera-Mesa**[17], **Diego Echeverry** [18], **Tim William**[1,19], **Nicholas M. Anstey**[1], **Matthew J. Grigg** [1], **Nicholas P. Day**[20,21],

**Nicholas J. White** ⓘ [20,21], **Dominic P. Kwiatkowski** ⓘ [9,23], **Aimee R. Taylor** ⓘ [22], **Rintis Noviyanti** [4], **Daniel Neafsey** ⓘ [5,6], **Ric N. Price** ⓘ [1,20,21] **& Sarah Auburn** ⓘ [1,20] ✉

---

[1]Menzies School of Health Research and Charles Darwin University, Darwin, NT, Australia. [2]Exeins Health Initiative, South Jakarta, Indonesia. [3]Global and Tropical Health Division, Menzies School of Health Research, Darwin, Australia. [4]Eijkman Molecular Biology Research Center, National Research and Innovation Agency, Cibinong, Indonesia. [5]Harvard T.H. Chan School of Public Health, Boston, MA, USA. [6]Broad Institute, Cambridge, MA, USA. [7]Australian Genome Research Facility, Brisbane, VIC, Australia. [8]Laboratory of Experimental Cardiology, Department of Cardiology, University Medical Center Utrecht, Utrecht, The Netherlands. [9]Wellcome Sanger Institute, Hinxton, UK. [10]Oxford University Clinical Research Unit, Hospital for Tropical Diseases, Ho Chi Minh City, Vietnam. [11]Ethiopian Public Health Institute, Addis Ababa, Ethiopia. [12]College of Medicine & Health Sciences, Arba Minch University, Arba Minch, Ethiopia. [13]Afghan International Islamic University, Kabul, Afghanistan. [14]Universitas Sumatera Utara, Medan, Indonesia. [15]Faculty of Medicine, University of Indonesia, Jakarta, Indonesia. [16]Infectious Diseases Division, International Centre for Diarrhoeal Disease Research, Bangladesh (icddr, b), Dhaka, Bangladesh. [17]Universidad de Antioquia, Medellin, Colombia. [18]Departamento de Microbiología, Facultad de Salud, Universidad del Valle, Cali, Colombia. [19]Queen Elizabeth Hospital, Kota Kinabalu, Malaysia. [20]Centre for Tropical Medicine and Global Health, Nuffield Department of Medicine, University of Oxford, Oxford, UK. [21]Mahidol-Oxford Tropical Medicine Research Unit, Mahidol University, Bangkok, Thailand. [22]Institut Pasteur, University de Paris, Infectious Disease Epidemiology and Analytics Unit, Paris, France. [23]Deceased: Dominic P. Kwiatkowski. [24]These authors contributed equally: Mariana Kleinecke, Edwin Sutanto.
✉e-mail: Sarah.Auburn@Menzies.edu.au

