## [Peer Review file · Nature Communications]

Microhaplotype deep sequencing assays to capture *Plasmodium vivax* infection lineages

Corresponding Author: Professor Sarah Auburn

Version 1:

Reviewer comments:

Reviewer #1

(Remarks to the Author)

Review of

Microhaplotype deep sequencing assays to capture *Plasmodium vivax* infection lineages

Summary: The authors present an interesting and well-prepared manuscript presenting a deep sequencing assay targeting 93 microhaplotypes, markers in 2 drug resistance genes and a plasmodium species detection marker. The microhaplotypes in the panel were selected from 2 previous panels published by the same team. The panel and assay were evaluated through two approaches: (1) in silico analysis using previously published whole genome sequencing data and (2) testing on samples from clinical and epidemiological trials conducted primarily in Afghanistan, Indonesia, Vietnam, and Ethiopia. This work is valuable to a broad scientific audience interested in malaria (molecular) surveillance of *P. vivax* malaria. The developed tool has the potential to significantly contribute to malaria surveillance and elimination programs, providing evidence to inform policy decisions, such as assessing re-infection in therapeutic efficacy studies.

Strengths: The manuscript is well written, with the results presented in a clear and comprehensible manner. The methods are appropriately documented, and the chosen approaches align well with the research objectives. The developed assay performs especially well in determining genetic relatedness (IBD), for which microhaplotype panels in general perform very well. Furthermore, the supplementary materials provide an in-depth description of the designed panel, and the manuscript is complemented by an online protocol. Additionally, the data processing pipeline is accessible through GitHub, enhancing transparency and supporting reproducibility and application of the tool.

Major Comments:

1. Assay Sensitivity and Parasite Density Thresholds: The manuscript aims to develop a surveillance tool suitable for implementation in *P. vivax*-endemic regions, emphasizing high sensitivity for low-density infections at an affordable cost. It is not clear what is considered as a low density infection by the authors, and what minimal density was aimed for or would be important for surveillance. In addition, the limit of detection is not clearly presented. Clarify the definition, presentation, and origin of these thresholds and specify if they apply similarly to whole blood and dried blood spot samples.
2. IBD Analysis and Panel Utility: The assay's 93 microhaplotypes offer an important advantage in the genotyping of recurrent infections using pairwise IBD analysis. More comprehensive presentation and details would enhance clarity:
 - a. Clarify how the Panel Judge estimator simulates data (line 201–207) and whether it is based on samples from this study. Explain why only four countries were assessed and if broader applicability using in silico WGS data is possible.
 - b. Summarize key findings from the Ethiopia trial for context. It might be important to summarize some main results from that study to help interpret the results. Note that the lower number of sample pairs in PQ arms is due to higher prevention of relapses, while more related pairs in non-PQ arms indeed indicate greater relapse risk. Address the basis for increased recrudescence risk in non-PQ arms, as this is currently speculative and these are challenging to differentiate from relapses through genetic relatedness alone.
 - c. Expand on the distribution of IBD scores across arms and present statistical comparisons for the distributions. The text (using frequencies) does not align well with the figure (showing distributions). Clarify how this helps distinguish relapse/reinfection from recrudescence, which is crucial for TES trial outcomes.
 - d. Justify the 120-day threshold for follow-up, which is longer than typical TES periods (e.g., 48 days). Explain the intent behind this duration.
 - e. Was any other genotyping method used in the clinical trial to assess recrudescences vs relapse/reinfection? If yes, how does the panel perform in comparison to the tool used in that study?
3. Drug Resistance Markers and Concordance Testing: The assay targets 4 resistance markers across 2 genes. The discussion notes a non-significant difference in prevalence relative to the baseline (lines 350–351). Present direct

comparisons between this assay's genotyping outcomes and those of other methods in the results section. Although WGS data were generated for validation, the concordance analysis with the NGS assay was not shown; this should be included.

a. Consider presenting cumulative haplotypes for each gene, as this would better reflect the phenotypic impact of mutations compared to individual marker analysis.

4. In the discussion, include a comparison of the new panel and assay with existing NGS panels for *P. vivax* surveillance. Currently, the manuscript only briefly compares it to the 42-SNP Broad panel genotyped using a high resolution melt assay. Expanding this comparison would provide more context on the panel's performance relative to other available NGS tools; e.g. Lin et al. 2015, Fola et al. 2020, Kattenberg et al. 2022, and optionally Popkin-hall 2024 (preprint MIPs assay).

Minor comments

- a. While the manuscript states that the assay detected "distinct separation of populations by national boundaries," it is important to note that the four countries analyzed are not geographically adjacent but widely separated.
- b. Line 65: Clarify the term "policy-relevant treatment," as all antimalarial treatments could be considered policy-relevant.
- c. Line 79: The panel and assay improve the ability to distinguish relapses from reinfections, but identifying identical relapses versus recrudescence remains challenging.
- d. Lines 99-101: Review the grammar and tense for accuracy.
- e. Line 115: change 'the multiplex' to 'the multiplex assay'
- f. Line 123/124: Specify the criteria used for deeming control samples permissible and state how many samples did not meet these criteria.
- g. Line 142: Clarify whether DBS or whole blood samples were used for the analysis.
- h. Line 151: Indicate the range of parasite densities
- i. Line 164: Replace "moderate proportion" with the exact proportion.
- j. Lines 158 – 166: are the markers mentioned here microhaplotype markers? The marker numbers are not very informative.
- k. Line 175-179: Include the difference in (mean/median) sequencing depth between WGS and the amplicon panel.
- l. Lines 185-188: Explain the impact of sequencing depth differences on discrepancies between eMOI and Fws.
- m. Line 193: Justify why only populations with 30 or more isolates were analyzed.
- n. Line 195: If something is not significant, then there is no trend. Please rephrase.
- o. Line 196: "The lowest eMOI distribution was observed in Sumatra, Indonesia,.. relative to the other sites" -> this is very hard to see in the figure with the current scale. Consider adding the proportion of polyclonal infections to illustrate the relationship with endemicity.
- p. Line 227 Revise "demonstrated differentiation between borders" to clarify, given that the countries are geographically distinct and do not share borders. Explain the phrase "with <50% probability of polyclonality," detailing if samples with >50% were excluded and how this was determined (e.g., using MOIRE). Add details to the methods section.
- q. Line 238: Compare the depth of the mitochondrial marker to nuclear targets and clarify its detection capability in mixed-species infections and the proportion threshold for minor species.
- r. Line 246 Note in the discussion that only dhps, not dhfr, was genotyped, limiting the ability to fully assess SP resistance.
- s. Line 367: Specify if the reported cost includes only library preparation or also quantification and sequencing. Include the total cost breakdown including these costs.
- t. Figure 1: Clarify the difference between the plots with 25 reads vs. 10 reads in the caption and help the reader to better interpret this figure. In addition, highlight regional performance variations of some markers, which may affect assay applicability across settings.
- u. Figure 2:
- a. Explain why Colombia and Bangladesh are excluded.
- b. Add that the data comes from 83 *P. vivax* cases with WGS data.
- v. Figure 5 c&d: Adjust the line thickness or color contrast for better distinction between blue and green lines.
- w. Figure 6: Ensure the caption and plot show consistent IBD thresholds (use 25, 50, and 95% if appropriate). Clarify why these specific thresholds were chosen.

(Remarks on code availability)

the github contains 2 *P. vivax* specific pipelines for data processing resulting in SNP calling and microhaplotype calling. The github includes documentation for installing and running the pipelines. It is not clearly described, but the pipeline seems specific for the amplicon panel described in the manuscript, and therefore has limited applicability to other panels. However, it does allow the reproduction of the analysis from the manuscript, and can be used by others who will use the same amplicon panel to process their raw data. The output description for the microhaplotype analysis has not yet been completed, but there are instructions how to run this analysis.

Reviewer #2

(Remarks to the Author)

This work reports the development, validation and application of a high throughput multiplex microhaplotyping assay to characterize individual *P. vivax* infections, building up on a recently published paper by the same group. The paper is well written and provides compelling evidence of how valid and useful this approach could be to characterize infections. The main result illustrating the power of this approach is the IBD measured for samples from the RCT analyzed with the clear impact of treatments with and without primaquine. Validation of variant calling using samples with available WGS data also provides compelling evidence of the accuracy of the method. Clearly, this approach has the potential to be applied widely in vivax research.

A comment I have is the limit of the approach to distinguish relapses from recrudescence of parasites. Although this is acknowledged by the authors in most of the manuscript, it should be more clearly emphasized. For example, in the abstract line 65, authors state that this method can inform policy-relevant treatment, I believe this should be toned down and with

validated use case explained. Similar suggestion for line 108. Besides, I am not convinced with the statements made line 257-260 about the prevalence of Y976F mutants in the Ethiopian RCT and I would suggest authors to tone down interpretation as, as it stands, recrudescence cannot be inferred to these recurrent infections.

I would suggest authors to test in the data the frequency of closely-related paired infections (at both >25% and >50%) in samples collected from different patients? How frequent is that, notably in Sumatra where diversity is lower than in other sites?

I think there is a mix-up in the abstract and higher frequency of relapse/recrudescence are in the no primaquine group, not the other way around.

The sensitivity analysis defined appropriate depth if more than 100 reads per marker were observed. However, I do not see an explanation for using this level of depth as appropriate for the analyses. Why not choosing 50 reads per marker? Or 200 reads? This should be better explained. Notably because line 152, successful genotyping is defined as >25 reads per marker.

Figure 2 c and 2d show analysis of eMOI across different geographic settings (country and province). Line 197 and in legend of Figure 2, authors state that differences in eMOI probably reflects differences in transmission intensity and lower eMOI are suggestive of low transmission. This seems indeed a rationale interpretation, but adding some kind of metrics of transmission in the different settings studied would help to strengthen this.

Having the same color code for the different countries in all panels of Figure 5 would be helpful.

The authors are quite clear in the manuscript of the uncertainty around the markers of drug resistance studied. However, they should consistently mention throughout the draft that these are candidate markers (example line 244)

(Remarks on code availability)

Reviewer #3

(Remarks to the Author)

Kleinecke et al. introduced a novel amplicon sequencing panel for *Plasmodium vivax*, demonstrating its application to IBD-based inferences of transmission patterns across samples from various geographic regions. The manuscript is well-written and provides important insights into the panel's performance in terms of sequencing coverage, genotype call accuracy, MOI inference, inference of IBD-based genetic relatedness, and its use for informing recurrence and population structure. However, the manuscript could be enhanced by addressing the following comments:

Major points:

1. The authors previously reported the design of the AmpSeq panel through in silico analysis, primarily relying on SNPs found in existing whole genome sequencing data without explicitly analyzing indels. Given the abundance of indels in the *P. vivax* genome (10.12688/wellcomeopenres.17795.1), it is possible that indels are enriched in the chosen amplicons. In this manuscript, biallelic SNPs (not indels) were used for the analysis involving WGS data when comparing WGS and AmpSeq. However, it is unclear whether indels were included when analyzing the AmpSeq data, including variant calling concordance analysis (Ln 168-179) and MOI estimation (Ln 181-190). If indels are used for AmpSeq data but not for WGS data, genotype call inconsistency and MOI may be overestimated. If so, I suggest the author perform a more matched analysis when comparing AmpSeq with WGS data by consistently including or excluding indels between AmpSeq and WGS data.

2. It is unclear how IBD-based genetic relatedness estimated from Dcifer helps in understanding recurrence, particularly relapse. Individuals in vivax-endemic regions can harbor a reservoir of genetically diverse hypozoites (10.1038/s41467-019-13412-x), not necessarily related to the blood-stage clone. High relatedness between initial and recurrent infections could indicate either recrudescence (from uncleared blood-stage clones) or relapse (from one of the hypozoite clones highly related to schizont clone), while low relatedness could suggest reinfection (unrelated clone) or relapse (from one of the hypozoites clones that is unrelated to schizont clone). Can the authors further clarify their claims between lines 209-221, particularly lines 218-221?

Besides, Dcifer is designed to measure inter-host genetic relatedness, assuming no within-host clone-clone level relatedness. The relatedness between pairs of clones from different hosts (instead of host-host pairs) is not individually measured; relatedness estimates at the host-host level may represent simplification or merging of clone-clone level relatedness into that of a host-host pair. As discerning different type of recurrence is conceptually at clone-clone level, relatedness estimated from Dcifer may complicate the interpretation how relatedness informs recurrence. I wonder whether it is feasible to perform a similar analysis focusing on a subset of monoclonal infections using hmmlBD to infer relatedness? This might simplify the interpretation of relatedness patterns. Comparing the relatedness patterns from Dcifer using all samples versus those derived from only monoclonal samples with hmmlBD could provide practical guidance for the research community on sample selection and choice of relatedness estimators.

Minor points:

3. In the "effective IBD capture" results section, the authors simulated five types of relationships, including $r = 0, 0.25, 0.5,$ and 0.75 . Figure 3 shows that the absolute error is greatest for intermediate r values. Although this indicates that IBD-based estimates of genetic relatedness are most problematic for intermediate r values like $r = 0.5$, the same error magnitude (e.g., RMSE = 0.05) can have varying implications depending on the true r value. For instance, with $r = 1.0$, an error of 0.05 amounts to 5% of the true relatedness, whereas, for $r = 0.25$, it represents 20% of the true r value, which is more concerning.

It might be more appropriate to present RMSE as a relative value by dividing by the true r value, especially for a population sample where most parasite pairs share very little IBD.

The simulated relationships, $r = 0, 0.25, 0.5, 0.75,$ and 1.0 , appear uniformly distributed; however, when converted to degrees of relatedness (10.1534/genetics.117.1122), the values $0.25, 0.5, 0.75,$ and 1.0 represent close relationships (from half-sibling to clones). This suggests that more distant relationships ($r = 0$ to $r = 0.25$) are largely unsampled in the simulation and therefore unevaluated. I recommend that the authors incorporate more values of r between 0 and 0.25 in the simulations to better reflect the distribution of genetic relatedness observed in the empirical samples.

4. In evaluating variant calling accuracy, the author suggested that discordant calls between WGS and AmpSeq data likely stem from differences in detecting minor clones at heterozygous sites, especially those with lower minor allele frequencies. However, the read depth for these sites in WGS and AmpSeq data is not provided. Is there a clear relationship between discordance in variant calls and the difference in allele read depth between WGS and AmpSeq? If analysis is restricted to sites and samples with equivalent read depth between WGS and AmpSeq data, will the genotype discordance rate be significantly reduced?

5. Studies (10.1534/genetics.110.113977 and 10.1038/s41467-024-46659-0) have demonstrated that strong directional selection, such as antimalarial drug pressure, can bias the IBD distribution and affect the accuracy of downstream analyses related to genetic relatedness. Consequently, including drug resistance markers might also skew estimates of genetic relatedness. I am interested in hearing how the authors plan to address these potential biases and whether these markers should be excluded when estimating genetic relatedness and population structure based on IBD.

6. Missing data is a common challenge in the genomic analysis of parasites and may be even more prevalent in AmpSeq due to its design for use with isolates with low DNA input or from low-quality dried blood spots. Although sensitivity is discussed in lines 131-140, it is unclear whether the read depth mentioned is averaged across markers or if each marker has at least a read depth of 100, for example. It is important to report the missingness of genotype calls, including the fraction of samples and markers filtered out during quality control steps, the number of samples and markers remaining for IBD-based analysis, the genotype call missing rate for the QC'ed samples and sites, and any recommendations for study designs to address this issue. For instance, should we consider increasing the sample size by a certain amount to confidently achieve the desired sample size for downstream analysis?

(Remarks on code availability)

Version 2:

Reviewer comments:

Reviewer #2

(Remarks to the Author)

Authors have addressed all points raised and I have no further comments.

(Remarks on code availability)

Reviewer #3

(Remarks to the Author)

The authors have substantially strengthened the manuscript in this round of revision and have addressed the majority of the points I raised as Reviewer #3. I have only a handful of follow-up questions and minor suggestions:

1. Incorporating clarifications into the main text

The response letter contains several helpful clarifications that would also benefit readers of the article itself. In particular, the explanation that "large relative errors at very low relatedness values are inconsequential when analyses focus on pairs above a threshold (e.g., relatedness ≈ 0.25)" could be distilled into one or two sentences in the Results or Discussion. Likewise, noting that studies targeting more distantly related isolates may require denser genotype data (e.g., whole-genome sequencing) would strengthen the discussion of limitations.

2. Definition of WGS / AmpSeq genotype discordance

The method used to compute discordance between WGS and AmpSeq genotypes remains unclear. If I understand correctly, the GATK-based WGS pipeline produces diploid genotype calls (e.g., A/A, A/T), thus limiting a site to only have a maximum of two alleles, whereas the AmpSeq pipeline can retain more than two alleles per SNP (e.g., A/T/G). If that interpretation is accurate, the discordance metric may be inflated by pipeline constraints in addition to sequencing depth. A brief clarification (perhaps in the Methods) of how multi-allelic sites are handled in each pipeline would resolve this concern.

3. Assumption in line 217

The assumption on line 217 should be explicitly framed as applying to low-inbreeding or high-transmission (high-

endemicity) populations. Adding that qualifier will prevent readers from over-generalizing the conclusion.

4. Interpretation of lines 248–249

The statement that “there was no evident differentiation by methodology, supporting the robustness of merging the data sources” warrants caution. Whole-genome sequencing, by virtue of its marker density, can detect more distant relatedness than targeted panels. Because the present analyses focus chiefly on isolate pairs with IBD > 0.25 -- where RMSE is low -- the claim of methodological equivalence may not extend to relationships below that threshold. A more nuanced phrasing acknowledging this limitation would be advisable.

(Remarks on code availability)

Response to Reviewers RE: Microhaplotype deep sequencing assays to capture *Plasmodium vivax* infection lineages

Please note, all line numbers in our responses refer to the track changes version of the manuscript.

Reviewer 1

We thank the reviewer for their thoughtful review. Before addressing specific comments, we'd like to address a potential misunderstanding: although we lead in the introduction with chloroquine resistance, for which the goal of classification analyses is separation of recrudescence from relapse/reinfection, in the section entitled "Potential... to inform on recurrence" we were motivated by a different use case: that of radical-cure efficacy estimation, for which the goal of classification analyses is separation of reinfection from relapse/recrudescence. We have updated the introduction to give greater emphasis on the importance and challenges of radical cure for *P. vivax* (see lines 77-94 as below) and have explicitly stated in line 124 that our results provide proof of concept in the assessment of radical-cure efficacy in a primaquine trial conducted in Ethiopia.

Lines 77-94: "Relapsing infections may enhance spatio-temporal *P. vivax* transmission and can cause repeated symptomatic illness, increasing the risk of anaemia and life-threatening disease 3. The radical cure of vivax malaria requires treatment with both a schizontocidal antimalarial (chloroquine (CQ) or artemisinin combination therapy) combined with a hypnozoiticidal agent (primaquine (PQ) or tafenoquine (TQ))¹. However, safe and effective treatment of hypnozoites is complicated by several factors. Both PQ and TQ can cause severe hemolysis in patients with a common enzymopathy, glucose-6-phosphate dehydrogenase (G6PD) deficiency, and hence healthcare providers may be hesitant to provide these treatments in the absence of point-of-care tests to diagnose the condition. The risk of hemolysis and the efficacy of PQ and TQ regimens are determined in part by the total dose and length of the treatment regimen and may vary in different endemic settings 4-8. Clinical trials of *P. vivax* with a long follow up period (generally 6-12 months) can provide critical insights on the risk of recurrence with different treatment regimens but are constrained in distinguishing whether recurrent infections are due to schizontocidal treatment failure (recrudescence), reactivation of hypnozoites (relapses), or a new mosquito inoculation (reinfection) 9,10."

Major comment 1. Assay Sensitivity and Parasite Density Thresholds: The manuscript aims to develop a surveillance tool suitable for implementation in *P. vivax*-endemic regions, emphasizing high sensitivity for low-density infections at an affordable cost. It is not clear what is considered as a low density infection by the authors, and what minimal density was aimed for or would be important for surveillance. In addition, the limit of detection is not clearly presented. Clarify the definition, presentation, and origin of these thresholds and specify if they apply similarly to whole blood and dried blood spot samples.

Our study was aimed primarily at developing genotyping tools for surveillance of symptomatic (clinical) patients. This is now clarified in line 119:

“Our assay was designed to primarily target clinical (symptomatic) cases.”

Defining *P. vivax* parasite density thresholds for symptomatic patients is complex but insights from a meta-analysis conducted by Groves et al suggest that our assay would successfully genotype >95% symptomatic cases. We have added the following comments to the discussion (lines 339-344):

“A critical requirement of the assay was its ability to be implemented in surveillance frameworks in vivax-endemic countries. This necessitated high sensitivity to genotype low-density, symptomatic *P. vivax* infections at affordable cost using locally accessible sequencing platforms. In a meta-analysis of 27 *P. vivax* clinical studies that did not apply a parasitemia threshold for enrolment, 95% of febrile patients had parasitemia >120 parasites per microliter blood²⁴. With full chemistry, our analyses of serial dilutions demonstrated that >80% genotyping was successful with yields of >100 mean read depth when parasite density was >70 per microliter whole blood. Hence, we estimate that >95% of symptomatic *P. vivax* cases should be successfully genotyped.”

Our serial dilutions were prepared from whole blood samples. We have generated new results derived from dried blood spots, which demonstrate high genotyping success across a range of parasite densities. The new results are illustrated in Supplementary Figure 6 and described in the results (lines 168-173):

“Our serial dilutions were prepared using blood collected into anticoagulant-coated tubes, that generally yield higher quality DNA than dried blood spots. We determined how the number of successfully genotyped markers correlated with parasitemia in dried blood spot samples from a study in Ethiopia. There was a weak correlation between parasitemia and read count ($\rho=0.1873$, $p=0.0217$); 148 cases (98.67%) with parasitemia ranging from 240-42,280 parasites/ul could be genotyped successfully at ≥ 80 (86%) microhaplotype markers (Supplementary Figure 6).”

Major comment 2. IBD Analysis and Panel Utility: The assay’s 93 microhaplotypes offer an important advantage in the genotyping of recurrent infections using pairwise IBD analysis. More comprehensive presentation and details would enhance clarity:

a. Clarify how the Panel Judge estimator simulates data (line 201–207) and whether it is based on samples from this study. Explain why only four countries were assessed and if broader applicability using *in silico* WGS data is possible.

Further information on the *paneljudge* simulation method has been added to the document on lines 600-607 (methods), including clarification that this was based on samples in this study:

“For each country, we estimated allele frequencies for the markers in the microhaplotype panel. These allele frequency estimates were then entered into *paneljudge* to evaluate the panel's per-country capacity to generate data for accurate relatedness estimation. In brief, for each r value evaluated, data on 100 parasite pairs were simulated under the hidden Markov model HMM model, with inter-marker distance computed by the difference between the amplicon (marker) midpoints. For each pair, simulated data were then used to estimate r (using *paneljudge* i.e., under the same model used to simulate the data). The RMSE of the 100 estimates around the data-simulating r value was then computed.”

We only used populations with a minimum of 20 samples to ensure that our allele frequency estimates were accurate. Having generated new data from Bangladesh and Colombia since the original submission, we now have 6 populations from different global regions.

We have previously shown *paneljudge* plots for microhaplotypes derived from WGS data (PMID: 39117628). To avoid overlap, we intentionally refrained from presenting a full WGS analysis again here.

b. Summarize key findings from the Ethiopia trial for context. It might be important to summarize some main results from that study to help interpret the results. Note that the lower number of sample pairs in PQ arms is due to higher prevention of relapses, while more related pairs in non-PQ arms indeed indicate greater relapse risk. Address the basis for increased recrudescence risk in non-PQ arms, as this is currently speculative and these are challenging to differentiate from relapses through genetic relatedness alone.

Thank you for this excellent suggestion. Within the discussion, we have added information on the main findings from the clinical analysis in the Ethiopian RCT (lines 386-390):

“The trial evaluated the safety and efficacy of combining PQ with either CQ or AL for the radical cure of *P. vivax*. There was a 2- to 3-fold reduction in the risk of recurrence within 12 months compared to patients treated with CQ or AL monotherapy, but there was no significant difference in the risk of recurrence between the CQ+PQ and AL+PQ arms.”

Since the original submission, we have generated additional data and now have 128 paired *P. vivax* isolates. These new data confirm that IBD is higher (more inferred recrudescence or relapse events) in the non-PQ than in the PQ treatment arms suggesting that PQ is highly effective at suppressing recrudescence/relapse. As detailed in the response to c) below, we have added more context on the low risk of recrudescence in this data set. An interesting new insight from our data that was not apparent from analysis of the clinical data alone, is that PQ+CQ appears to have greater efficacy to suppress recrudescence/relapse events than PQ+AL. However, this did not reach statistical significance due to relatively small sample size.

As outlined in the discussion (lines 388-394), we now provide some comments on features of the drugs that may explain this interesting trend, we also highlight the utility of genotyping to provide greater insights into clinical trial data:

“The genetic data provided additional insights to interpret the clinical trial. Whilst there was no overall difference in the risk of recurrence between patients treated with PQ+CQ and PQ+AL, PQ+CQ reduced the proportion of highly related pairs compared to PQ+AL. This result may reflect CQ's slower elimination and longer post-treatment prophylaxis compared to AL, that suppressed early relapses 31. Furthermore, CQ has been shown to have synergistic activity with PQ, that is not apparent with AL, and this may have had greater suppression of relapses and related recurrences 32. However, the difference between CQ+PQ and AL+PQ did not reach significance: further studies are warranted to explore this.”

c. Expand on the distribution of IBD scores across arms and present statistical comparisons for the distributions. The text (using frequencies) does not align well with the figure (showing distributions). Clarify how this helps distinguish relapse/reinfection from recrudescence, which is crucial for TES trial outcomes.

Regarding the comment on TES trial outcomes, please see our clarification in the opening statement for this response.

Regarding the IBD distribution, we chose to present these distributions as this gives a more complete picture than showing frequencies at pre-defined IBD thresholds. We have added statistical comparisons on the IBD distributions by treatment arm (lines 260-262, 265-266 and 270-272) but have retained some statistics on the frequencies as we anticipate some readers may find this more intuitive.

We used IBD distributions and thresholds to inform on probable relapse versus reinfection risks in each treatment arm under the assumption that highly related pairs are more likely to reflect relapses than reinfections. Some highly related pairs may reflect recrudescence; the risks of recrudescence need to be interpreted in the context of the study setting. In the future, recrudescence should be further distinguishable from relapse using time-to-event models and models that differentiate between inter-episode sibling relationships (compatible with relapse but not recrudescence) and inter-episode clonal relationship (compatible with both relapse and recrudescence). We have amended the text to clarify these points as outlined below.

- Within lines 257-259, we have added clarification on how IBD data were used in the study: “Using the full set of 128 microhaplotype-genotyped pairs from the RCT, we used IBD distributions and thresholds to inform on probable relapse/recrudescence versus reinfection risks in each treatment arm under the assumption that highly related pairs are more likely to reflect relapses than reinfections.”
- We have added more context on the risks of recrudescence for CQ and AL in the discussion (lines 392-400): “Some of the highly related pairs may reflect recrudescence but we postulate that the majority reflect relapses in the absence of radical cure. At the time of the study (2012-14), there was no local evidence for AL resistance even in *P. falciparum*. There was evidence of low-grade CQ resistance in *P. vivax*, although only 4% of the genotyped recurrences in the study occurred before day 28. The integration of genetic models of clonal versus inter-episode sibling relationships with time-to-event analyses will eventually support further dissection of relapses from recrudescence 30.”

d. Justify the 120-day threshold for follow-up, which is longer than typical TES periods (e.g., 48 days). Explain the intent behind this duration.

Please see our clarification on TES versus radical-cure efficacy trials in the opening statement for this reviewer response. The data presented from the RCT in Ethiopia is from a clinical trial of radical cure treatments, which generally have prolonged follow up (6-12 months) to capture late relapses. The short follow-up adopted in TES is not ideal to determine antirelapse efficacy. Relapses in most tropical locations occur early and the outcome at 120 days is a suitable compromise between early relapses and later reinfections (PMID: 24731298); this is now clarified in the text (lines 279-281).

e. Was any other genotyping method used in the clinical trial to assess recrudescences vs relapse/reinfection? If yes, how does the panel perform in comparison to the tool used in that study?

A primary objective of our markers was to capture IBD accurately so that sibling relationships could be detected between different time points - this approach gives strong evidence of relapse as opposed to reinfection. The original Ethiopian RCT evaluation produced

microsatellite genotyping data on 46 pairs of infections where the recurrence occurred at or before day 42. With only 1-7 microsatellites, we decided not to measure IBD, so their classifications were constrained to “heterologous” or “homologous” definitions. We compared the microsatellite classifications from 34 pairs of infections that have corresponding microhaplotype data which demonstrated an overestimation of strangers with the microsatellite data. Details on the comparison are presented in lines 252-255 including:

“Amongst the 34 pairs, 44% (15/34) were defined as heterologous using the microsatellite data compared to 18% (6/34) defined as strangers (arbitrary threshold of IBD <25%) with the microhaplotype-based IBD estimate (Supplementary Table 6). The relative overestimation of strangers with microsatellite data underscores the improved insight from IBD analysis.”

Major Comment 3. Drug Resistance Markers and Concordance Testing: The assay targets 4 resistance markers across 2 genes. The discussion notes a non-significant difference in prevalence relative to the baseline (lines 350–351). Present direct comparisons between this assay’s genotyping outcomes and those of other methods in the results section. Although WGS data were generated for validation, the concordance analysis with the NGS assay was not shown; this should be included.

The concordance analysis between WGS and amplicon sequencing data includes all microhaplotype and drug resistance markers; this has been clarified in the text on line 200. As described in the text on line 201 there was overall very high concordance (>96%) between the methods.

Please also note that none of the samples in the WGS validation came from the Ethiopian RCT. Attempts to obtain WGS data from the RCT samples were not very successful, hence the need for this targeted genotyping effort. The majority of Ethiopian WGS samples described here derive from another study that involved a 28-day TES of CQ (clinical data unpublished).

a. Consider presenting cumulative haplotypes for each gene, as this would better reflect the phenotypic impact of mutations compared to individual marker analysis.

We are conscious that these are drug resistance candidates (there are no validated drug resistance markers for *P. vivax*) as we emphasize in the discussion and as raised by reviewer 2 (comment 5). Furthermore, there are only a few markers for each gene. We have therefore chosen not to divulge into haplotype-based analyses as there is little value-add and this rather detracts from other, more important messages in the study.

Major Comment 4. In the discussion, include a comparison of the new panel and assay with existing NGS panels for *P. vivax* surveillance. Currently, the manuscript only briefly compares it to the 42-SNP Broad panel genotyped using a high resolution melt assay. Expanding this comparison would provide more context on the panel's performance relative to other available NGS tools; e.g. Lin et al. 2015, Fola et al. 2020, Kattenberg et al. 2022, and optionally Popkin-hall 2024 (preprint MIPs assay).

A discussion on utilities of our microhaplotype panel relative to other existing *P. vivax* barcodes have been added to the discussion (lines 482-490):

“Other targeted *P. vivax* genotyping assays are available, with designs tailored to different use cases 52. Several barcodes with amplicons targeting 40-200 single SNPs have been designed to capture infection diversity and inform on transmission but were not specifically designed to measure IBD, which requires consideration of both marker polymorphism and spacing 28,53,54. Most panels have been designed for amplicon sequencing but a recent panel of over 1,200 SNP targets was designed for genotyping using molecular inversion probes (MIPs) 52. The MIP panel was not evaluated for IBD determination, but the large number of markers brings potential. However, large marker numbers also confer relatively lower per marker sequence depth and less flexibility in sample throughput. MIP typing is not yet being used in malaria-endemic countries where amplicon sequencing offers a nimbler tool for surveillance.”

Minor comments

a. While the manuscript states that the assay detected “distinct separation of populations by national boundaries,” it is important to note that the four countries analyzed are not geographically adjacent but widely separated.

This is a good point. We have integrated WGS data at the marker positions to obtain data from 15 countries and prepared an NJ tree (new Figure 5), which illustrates high differentiation of countries by their national borders.

Note that some countries such as Vietnam and Cambodia have little resolution even with full WGS data owing to very high gene flow across borders. But also note that in some settings we even see moderate sub-national separation (see Vietnam and Ethiopia provincial networks in Figure 6).

b. Line 65: Clarify the term “policy-relevant treatment,” as all antimalarial treatments could be considered policy-relevant.

We have removed “policy-relevant”.

c. Line 79: The panel and assay improve the ability to distinguish relapses from reinfections, but identifying identical relapses versus recrudescence remains challenging.

Apologies, we are not sure what is requested here but we have been sure to highlight the challenge of recrudescence – please see response to major 2c above.

d. Lines 99-101: Review the grammar and tense for accuracy.

Amended “captured” to “capture”

e. Line 115: change ‘the multiplex’ to ‘the multiplex assay’

Amended

f. Line 123/124: Specify the criteria used for deeming control samples permissible and state how many samples did not meet these criteria.

Apologies for confusion, we have removed this line as the more important filters were applied using coverage and depth information.

g. Line 142: Clarify whether DBS or whole blood samples were used for the analysis.

Whole blood samples are now clarified in the text on **line 168** where we describe data from DBS.

h. Line 151: Indicate the range of parasite densities

Parasite densities (where available) are now provided in Supplementary Table 2.

i. Line 164: Replace “moderate proportion” with the exact proportion.

Updated the line to clarify this is a relative drop in read-pair versus read amount counts.

j. Lines 158 – 166: are the markers mentioned here microhaplotype markers? The marker numbers are not very informative.

These are microhaplotypes as now clarified in **line 189**. We opted for concise labels rather than a long label including e.g. chromosome and position details for every SNP within the amplicon. However, these details are provided for every marker in Supplementary Table 1.

k. Line 175-179: Include the difference in (mean/median) sequencing depth between WGS and the amplicon panel.

For conciseness, we have retained the text as it is since the doubling of heterozygous amplicon sequencing calls that were homozygous in the WGS dataset is a very clear sign of a minor allele genotyping influence.

l. Lines 185-188: Explain the impact of sequencing depth differences on discrepancies between eMOI and Fws.

We did not explore this in any further details as the correlation between eMOI and WGS with the new data (increased from 83 to 104 WGS samples) was very high ($\rho = -0.6109985$, $p = 5.664e-12$).

m. Line 193: Justify why only populations with 30 or more isolates were analyzed.

When sample size is low, minor allele frequency estimates (which underlie most population genetic measures) are inaccurate. At 20-30 or more samples, allele frequencies start to stabilize.

n. Line 195: If something is not significant, then there is no trend. Please rephrase.

Removed the word “trend”

o. Line 196: “The lowest eMOI distribution was observed in Sumatra, Indonesia,.. relative to the other sites” -> this is very hard to see in the figure with the current scale. Consider adding the proportion of polyclonal infections to illustrate the relationship with endemicity.

Labels indicating the percentage of polyclonal infections have been added to the plots.

p. Line 227 Revise “demonstrated differentiation between borders” to clarify, given that the countries are geographically distinct and do not share borders. Explain the phrase “with <50% probability of polyclonality,” detailing if samples with >50% were excluded and how this was determined (e.g., using MOIRE). Add details to the methods section.

With the new data derived from WGS samples, we now show excellent differentiation of national borders aside from regions such as the Vietnam/Cambodia border where resolution is not even possible with genomic data (new Figure 5). Details on polyclonality definitions and MOIRE methodology have been added to the Methods (lines 592-595).

q. Line 238: Compare the depth of the mitochondrial marker to nuclear targets and clarify its detection capability in mixed-species infections and the proportion threshold for minor species.

We have new data demonstrating capacity to detect both species in mixed infections (see Supplementary Table 4). We have not tested the minor limit of species detection.

r. Line 246 Note in the discussion that only dhps, not dhfr, was genotyped, limiting the ability to fully assess SP resistance.

Note added on line 326.

s. Line 367: Specify if the reported cost includes only library preparation or also quantification and sequencing. Include the total cost breakdown including these costs.

We have updated the costing to show the cost for the library preparation kits at \$AUD 13-15 per sample (lines 490-492). We did not provide sequencing costs as this is highly variable (and hence uninformative) depending upon the platform used, multiplexing level and other factors.

t. Figure 1: Clarify the difference between the plots with 25 reads vs. 10 reads in the caption and help the reader to better interpret this figure. In addition, highlight regional performance variations of some markers, which may affect assay applicability across settings.

We have updated the legend to clarify that one plot shows read amount and the other shows pairs (i.e. overlapping read pairs).

u. Figure 2:

a. Explain why Colombia and Bangladesh are excluded.

Colombia and Bangladesh previously had small sample size but this has since been improved and they are now included.

b. Add that the data comes from 83 *P. vivax* cases with WGS data.

This was previously noted in the legend, but we have adjusted the text to try to make it even clearer (line 912).

v. Figure 5 c&d: Adjust the line thickness or color contrast for better distinction between blue and green lines.

Figure and lines are now updated.

w. Figure 6: Ensure the caption and plot show consistent IBD thresholds (use 25, 50, and 95% if appropriate). Clarify why these specific thresholds were chosen.

We now have a new network plot design that enables several thresholds to be viewed in a single plot.

Remark 1 (Remarks on code availability):

The github contains 2 *P. vivax* specific pipelines for data processing resulting in SNP calling and microhaplotype calling. The github includes documentation for installing and running the pipelines. It is not clearly described, but the pipeline seems specific for the amplicon panel described in the manuscript, and therefore has limited applicability to other panels.

We have updated our pipeline to be more generic. The latest pipeline has built-in support for other barcodes as demonstrated for the *P. falciparum* SpotMalaria panel. We have also updated the documentation to include instructions on how to add custom panels. Please see updates on our GitHub site at <https://github.com/vivaxgen/MicroHaps>.

Reviewer 2.

Major comment 1. One comment I have is the limit of the approach to distinguish relapses from recrudescence of parasites. Although this is acknowledged by the authors in most of the manuscript, it should be more clearly emphasized. For example, in the abstract line 65, authors state that this method can inform policy-relevant treatment, I believe this should be toned down and with validated use case explained. Similar suggestion for line 108.

We have adjusted the text to further clarify that IBD has potential to improve recurrence classification, particularly when combined with time-to-event data, but may indeed be imperfect. Examples include:

- The abstract (lines 67-69) now reads "Our results demonstrate the potential to derive new information on *P. vivax* treatment and transmission using IBD generated by amplicon sequencing data that can be further improved with time-to-event models."
- The introduction (lines 121-125) now reads "Standard population-level genetic metrics could be applied to the data to highlight potential use cases for National Malaria Control Programs (NMCPs) that could inform *P. vivax* treatment options and transmission dynamics."

Proof of concept was provided in the assessment of radical-cure efficacy in a primaquine trial conducted in Ethiopia."

Regarding use cases, as detailed in response to reviewer 1 major comment 2c, with new data within the Ethiopian RCT, we demonstrate an observation on treatment efficacy from the genetic data that was not detectable with the clinical data alone. Specifically, we reveal a greater impact of PQ in reducing highly related pairs when paired with CQ than when paired with AL. This result can be readily explained by the slower elimination of CQ relative to AL, and differing interactions between CQ and PQ. Versus AL and PQ. This demonstrated insight on treatment is now described in lines 401-407.

Besides, I am not convinced with the statements made line 257-260 about the prevalence of Y976F mutants in the Ethiopian RCT and I would suggest authors to tone down interpretation as, as it stands, recrudescence cannot be inferred to these recurrent infections. Clarify further on why further modelling is needed and being worked on and tone down the paragraph.

We agree that the Ethiopian RCT is not the ideal study to investigate candidate determinants of CQ resistance as the number of recurrences by day 28 was very low (4% in the clinical dataset) and does not permit a suitably powered analysis. Furthermore, pvmdr1 is not a validated marker of CQ resistance. We have therefore removed the pvmdr1 analysis within the RCT to make space for greater detail on the results and discussion of the IBD analysis within the RCT.

We have added more details on model developments that are needed to further improve recurrence classification approaches integrating genetics (lines 410-424): "Our findings contribute proof of concept on the utility of genetics to inform on recurrence and highlight the potential insights that can be gleaned from recurrence classification of clinical trials. However, there are areas needing further research or development. Data from known relapse events, i.e. from individuals who were relocated to non-endemic areas, will be needed to confirm these findings. Further work is also needed to dissect the potential contribution of the splenic reservoir to recurrent *P. vivax* infections. Lastly, as well as integrating time-to-event information and resolving inter-sibling and inter-clone parasites, modeling approaches that include information on endemicity as a prior will be important. For example, the high inbreeding in Sumatra would complicate distinction of relapses from reinfections in this low endemic setting where many infections from independent patients are highly related. However, with low endemicity, it also follows that the risk of reinfection in Sumatra should be low. Our data have been made open access to support further modeling developments."

Major comment 2. I would suggest authors to test in the data the frequency of closely-related paired infections (at both >25% and >50%) in samples collected from different patients? How frequent is that, notably in Sumatra where diversity is lower than in other sites?

This is an excellent point. In some very low-endemic populations such as Sumatra, inbreeding may be high and independent infections are frequently highly related. In these settings, it is difficult to distinguish reinfections from relapses using genetic data alone. It should, however, also be noted that, if endemicity is low, it follows that reinfection should be infrequent and hence a less probable cause of recurrences. We have now clarified this dynamic in the

discussion and also highlighted the need for models that incorporate endemicity as a prior (lines 418-421).

I think there is a mix-up in the abstract and higher frequency of relapse/recrudescence are in the no primaquine group, not the other way around.

Thank you for pointing out the mix-up, this has now been resolved.

Major comment 3. The sensitivity analysis defined appropriate depth if more than 100 reads per marker were observed. However, I do not see an explanation for using this level of depth as appropriate for the analyses. Why not choosing 50 reads per marker? Or 200 reads? This should be better explained. Notably because line 152, successful genotyping is defined as >25 reads per marker.

Apologies for the confusion here. As clarified in line 177, for the current study, "Successful genotyping was defined as ≥ 25 reads for >80% of the 97 nuclear genome amplicons". The 25 read threshold was based on our specificity analysis in non-vivax samples (please see Supplementary Figure 2) but we note that other data sets may need different thresholds. For consistency, we have omitted all references to other thresholds and now only report on the 25 read and 80% markers genotyped thresholds in Supplementary Figures 4, 5 and 6. Our updated serial dilution figure (Supplementary Figure 2) is also now designed to allow the reader to evaluate the data at the 25 read or other chosen threshold.

Major comment 4. Figure 2 c and 2d show analysis of eMOI across different geographic settings (country and province). Line 197 and in legend of Figure 2, authors state that differences in eMOI probably reflects differences in transmission intensity and lower eMOI are suggestive of low transmission. This seems indeed a rationale interpretation, but adding some kind of metrics of transmission in the different settings studied would help to strengthen this.

Having the same color code for the different countries in all panels of Figure 5 would be helpful.

We initially considered including data on endemicity. However, as described in lines 438-439, the transmission dynamics reflected in the observed eMOI and IBD results may not be readily captured by standard measures of parasite incidence as relapses are a major contributor to the *P. vivax* burden.

We have updated the figures to include consistent colors for the 6 major populations.

Comment 5: The authors are quite clear in the manuscript of the uncertainty around the markers of drug resistance studied. However, they should consistently mention throughout the draft that these are candidate markers (example line 244)

Thank you, we have updated line 321 to clarify that the drug resistance-related targets are

indeed candidate markers.

Reviewer 3

Major comment 1. The authors previously reported the design of the AmpSeq panel through in silico analysis, primarily relying on SNPs found in existing whole genome sequencing data without explicitly analyzing indels. Given the abundance of indels in the *P. vivax* genome (10.12688/wellcomeopenres.17795.1), it is possible that indels are enriched in the chosen amplicons. In this manuscript, biallelic SNPs (not indels) were used for the analysis involving WGS data when comparing WGS and AmpSeq. However, it is unclear whether indels were included when analyzing the AmpSeq data, including variant calling concordance analysis (Ln 168-179) and MOI estimation (Ln 181-190). If indels are used for AmpSeq data but not for WGS data, genotype call inconsistency and MOI may be overestimated. If so, I suggest the author perform a more matched analysis when comparing AmpSeq with WGS data by consistently including or excluding indels between AmpSeq and WGS data.

This is a good point. We have included indel filtration to the pipeline and redone all the variant calling and all downstream population genetic analyses with indels removed.

Major comment 2. It is unclear how IBD-based genetic relatedness estimated from Dcifer helps in understanding recurrence, particularly relapse. Individuals in vivax-endemic regions can harbor a reservoir of genetically diverse hypnozoites (10.1038/s41467-019-13412-x), not necessarily related to the blood-stage clone. High relatedness between initial and recurrent infections could indicate either recrudescence (from uncleared blood-stage clones) or relapse (from one of the hypozoite clones highly related to schizont clone), while low relatedness could suggest reinfection (unrelated clone) or relapse (from one of the hypozoites clones that is unrelated to schizont clone). Can the authors further clarify their claims between lines 209-221, particularly lines 218-221?

We have addressed some of these comments in our reply to **reviewer 1 major 2b and reviewer 2 major 1**, including updates to the text. In brief, genetics alone cannot distinguish all relapses, but the IBD approach improves distinction of sibling-related relapses / recrudescence from less related relapses / reinfections. Whilst not perfect, this provides a more nuanced assignment of relapse than would be permissible with simple heterologous/homologous classifications. As we clarify in the discussion (**lines 415-424**), further developments are still needed. These include models that combine the genetics with time-to-event information and endemicity priors to further improve classification. What our assay brings for the first time is the ability to capture relationships with sibling-level relatedness with a parsimonious panel i.e., without needing whole genome sequencing. Our analysis of the RCT data in Ethiopia, including the demonstration of different impacts of PQ when combined with CQ versus AL that was not previously detected with the clinical data alone, provides a strong proof of concept for the utility of integrating IBD-based analyses into *P. vivax* clinical trial interpretation.

Besides, Dcifer is designed to measure inter-host genetic relatedness, assuming no within-host clone-clone level relatedness. The relatedness between pairs of clones from different hosts (instead of host-host pairs) is not individually measured; relatedness estimates at the host-host level may represent simplification or merging of clone-clone level relatedness into that of a host-host pair. As discerning different type of recurrence is conceptually at clone-clone level, relatedness estimated from Dcifer may complicate the interpretation how relatedness informs recurrence. I wonder whether it is feasible to perform a similar analysis focusing on a subset of monoclonal infections using hmmlBD to infer relatedness? This might simplify the interpretation of relatedness patterns. Comparing the relatedness patterns from Dcifer using all samples versus those derived from only monoclonal samples with hmmlBD could provide practical guidance for the research community on sample selection and choice of relatedness estimators.

Thank you for this suggestion. We agree that recurrence classification ideally requires a clone-to-clone level approach (the focus of statistical genetic models currently under development) and that the intra-host independence assumption of Dcifer undoubtedly leads to misspecification. This misspecification complicates the interpretation of individual relatedness estimates. In our recurrence analyses we are using Dcifer estimates comparatively: to compare IBD distributions across treatment groups. For a comparative analysis, what matters most is that we have a convenient measure of genetic distance / proximity (e.g. single value) that is well resolved (e.g. maps onto the full 0 to 1 range, unlike IBS sharing). Both Dcifer and hmmlBD provide a convenient and well-resolved measure of genetic proximity. Dcifer has the added advantage of providing a measure when samples are polyclonal. As shown below, Dcifer and hmmlBD are in good agreement for monoclonal samples.

We conducted a comparison of DCifer versus hmmlBD using the monoclonal infections from each of the 6 major populations, which reflect a range of endemic settings. The analyses were conducted using a maximum of 1000 fit iterations and number of iterations, with all other parameters at default. Additionally, to accommodate the limitation inherent to hmmlBD, we converted its input data to the major genotype. As illustrated in the figure below, we observed high concordance in the IBD estimates in all six populations (Spearman's rank correlation analysis, ρ range 0.47-0.97, all $p > 0.05$). Whilst we agree that a detailed description of different IBD measures is useful, it is beyond the remit of the current study and hence we have not included the plot below in the manuscript; this may be more appropriate in a separate study focused on sample selection and relatedness measures.

Dcifer IBD vs hmmIBD - Afghanistan

Dcifer IBD vs hmmIBD - Bangladesh

Dcifer IBD vs hmmIBD - Colombia

Dcifer IBD vs hmmIBD - Ethiopia

Dcifer IBD vs hmmIBD - Indonesia

Dcifer IBD vs hmmIBD - Vietnam

Major comment 3. In the "effective IBD capture" results section, the authors simulated five

types of relationships, including $r = 0, 0.25, 0.5,$ and 0.75 . Figure 3 shows that the absolute error is greatest for intermediate r values. Although this indicates that IBD-based estimates of genetic relatedness are most problematic for intermediate r values like $r = 0.5$, the same error magnitude (e.g., $RMSE = 0.05$) can have varying implications depending on the true r value. For instance, with $r = 1.0$, an error of 0.05 amounts to 5% of the true relatedness, whereas, for $r = 0.25$, it represents 20% of the true r value, which is more concerning. It might be more appropriate to present RMSE as a relative value by dividing by the true r value, especially for a population sample where most parasite pairs share very little IBD.

The simulated relationships, $r = 0, 0.25, 0.5, 0.75,$ and 1.0 , appear uniformly distributed; however, when converted to degrees of relatedness (10.1534/genetics.117.1122), the values $0.25, 0.5, 0.75,$ and 1.0 represent close relationships (from half-sibling to clones). This suggests that more distant relationships ($r = 0$ to $r = 0.25$) are largely unsampled in the simulation and therefore unevaluated. I recommend that the authors incorporate more values of r between 0 and 0.25 in the simulations to better reflect the distribution of genetic relatedness observed in the empirical samples.

Thank you for the suggestions. We re-ran *paneljudge* with a greater range of r values between 0 and 0.25; specifically adding in $r = 0.05, 0.1, 0.15,$ and 0.20 . The RMSE values for the new r values remained under 0.12, with the highest RMSE still observed at $r=0.25$ or $r=0.5$ (updated Figure 3).

We also explored relative RMSE values as illustrated in the figure below. Relative error decreases rapidly with increasing relatedness. Large relative errors at low relatedness values are not practically important because recurrence inference and most population genetic analyses focus on pairs above a threshold. Otherwise stated, although relative errors are large close to zero, they are small in absolute terms, and we can afford to make small errors at low relatedness values providing the estimate remains below the threshold. For example, the 100% error of an estimate of 0.02 of a true r value of 0.01 has no bearing on the proportion of pairs above 0.25. Moreover, relative errors hide the relative difficulty of accurately estimating intermediate r values. We therefore decided not to include the relative RMSE in our manuscript.

Major comment 4. In evaluating variant calling accuracy, the author suggested that discordant calls between WGS and AmpSeq data likely stem from differences in detecting minor clones at heterozygous sites, especially those with lower minor allele frequencies. However, the read depth for these sites in WGS and AmpSeq data is not provided. Is there a clear relationship between discordance in variant calls and the difference in allele read depth between WGS and AmpSeq? If analysis is restricted to sites and samples with equivalent read depth between WGS and AmpSeq data, will the genotype discordance rate be significantly reduced?

The variant calling criteria has been specified in supplementary Table 2 & 3. We have now provided read depth details for the genotypes used in the concordance analysis and identified the allele depth for WGS and AmpSeq genotypes at discordant genotypes (new Supplementary Figure 7). The results demonstrate that majority of the discordant genotypes are observed in cases with higher microhaplotype than WGS read counts.

Major comment 5. Studies (10.1534/genetics.110.113977 and 10.1038/s41467-024-46659-0) have demonstrated that strong directional selection, such as antimalarial drug pressure, can bias the IBD distribution and affect the accuracy of downstream analyses related to genetic relatedness. Consequently, including drug resistance markers might also skew estimates of genetic relatedness. I am interested in hearing how the authors plan to address these potential biases and whether these markers should be excluded when estimating genetic relatedness and population structure based on IBD.

This is a good point. As is generally best practice for malaria molecular surveillance studies, we excluded the drug resistance markers from our IBD-based analyses. For clarity, we have noted the markers used in each analysis in Supplementary Figure 2.

Major comment 6. Missing data is a common challenge in the genomic analysis of

parasites and may be even more prevalent in AmpSeq due to its design for use with isolates with low DNA input or from low-quality dried blood spots. Although sensitivity is discussed in lines 131-140, it is unclear whether the read depth mentioned is averaged across markers or if each marker has at least a read depth of 100, for example. It is important to report the missingness of genotype calls, including the fraction of samples and markers filtered out during quality control steps, the number of samples and markers remaining for IBD-based analysis, the genotype call missing rate for the QC'ed samples and sites, and any recommendations for study designs to address this issue. For instance, should we consider increasing the sample size by a certain amount to confidently achieve the desired sample size for downstream analysis?

This query has some overlap with **reviewer 2 major comment 3**. We have updated all results (**Supplementary Figures 4, 5 and 6**) referring to sensitivity to now use the threshold of ≥ 25 reads for $>80\%$ of the 97 nuclear genome amplicons to consider successful genotyping for a given sample. Our updated serial dilution figure (**Supplementary Figure 2**) is also now designed to allow the reader to evaluate the data at the 25 read or other chosen threshold. Our threshold is based on evaluation of negative controls in our data, but other data generated on other sequencing platforms (with higher or lower yield) may need different thresholds. We note some of the challenges in defining meaningful sensitivity thresholds and areas where variability in read depth can arise within the discussion (**lines 356-361**).

Response to reviewers

Reviewer #3 (Remarks to the Author):

The authors have substantially strengthened the manuscript in this round of revision and have addressed the majority of the points I raised as Reviewer #3. I have only a handful of follow-up questions and minor suggestions:

1. Incorporating clarifications into the main text

The response letter contains several helpful clarifications that would also benefit readers of the article itself. In particular, the explanation that “large relative errors at very low relatedness values are inconsequential when analyses focus on pairs above a threshold (e.g., relatedness ≈ 0.25)” could be distilled into one or two sentences in the Results or Discussion. Likewise, noting that studies targeting more distantly related isolates may require denser genotype data (e.g., whole-genome sequencing) would strengthen the discussion of limitations.

We have added a sentence to the main text describing why we chose not to use relative errors (Methods, lines 529 -531).

2. Definition of WGS / AmpSeq genotype discordance

The method used to compute discordance between WGS and AmpSeq genotypes remains unclear. If I understand correctly, the GATK-based WGS pipeline produces diploid genotype calls (e.g., A/A, A/T), thus limiting a site to only have a maximum of two alleles, whereas the AmpSeq pipeline can retain more than two alleles per SNP (e.g., A/T/G). If that interpretation is accurate, the discordance metric may be inflated by pipeline constraints in addition to sequencing depth. A brief clarification (perhaps in the Methods) of how multi-allelic sites are handled in each pipeline would resolve this concern.

We suspect the author may have missed that only individual SNP calls were used for the discordance analysis. The article already states that this analysis was based on the SNPs within the microhaplotypes on line 167 “was confirmed at the 425 SNPs within the 97 markers”. To remove any doubt, we have further clarified by added “SNP” again on line 165.

3. Assumption in line 217

The assumption on line 217 should be explicitly framed as applying to low-inbreeding or high-transmission (high-endemicity) populations. Adding that qualifier will prevent readers from over-generalizing the conclusion.

We have added a note to line 218, which now reads “under the assumption that highly related pairs are more likely to reflect relapses than reinfections (assumption suitable for low-inbreeding populations)”. Please also note that the discussion (lines 357-361) already describes the challenges of this assumption in low endemicity populations.

4. Interpretation of lines 248 -249

The statement that “there was no evident differentiation by methodology, supporting the robustness of merging the data sources” warrants caution. Whole-genome sequencing, by virtue of its marker density, can detect more distant relatedness than targeted panels. Because the present analyses focus chiefly on isolate pairs with IBD > 0.25 -- where RMSE is low -- the claim of methodological equivalence may not extend to relationships below that threshold. A more nuanced phrasing acknowledging this limitation would be advisable.

We suspect the author may have missed that this analysis doesn’t use the whole genome, as that would have left many missing calls in the amplicon sequencing data – it only uses data at the microhaplotype marker positions. We have further clarified this in lines 866-868, which now reads “The plot presents an unrooted neighbour-joining tree derived from a distance matrix on the microhaplotype calls using genotypes derived from microhaplotype and WGS data at the microhaplotype marker positions only”.